He *et al. Genome Biology*    (2025) 26:158

**METHODOLOGY**

# Integration of single-cell and spatial transcriptomics by SEU-TCA reveals the spatial origin of early cardiac progenitors

Jingjing He[1†], Yi Yang[1,2†], Rui Jiang[3†], Yanying Zheng[1†], Xianfa Yang[4], Xu Jiang[1], Xin Xue[1], Zhongzhou Yang[5], Naihe Jing[4], Hailong Cao[1*], Zhuojuan Luo[1*], Ke Wei[3*], Peng Xie[6*] and Chengqi Lin[1,2*]

†Jingjing He, Yi Yang, Rui Jiang and Yanying Zheng contributed equally to this work.

*Correspondence:
shuqu_1982@sina.com;
zjluo@seu.edu.cn; kewei@tongji.edu.cn; pengx@seu.edu.cn;
cqlin@seu.edu.cn

[1] Department of Cardiac Surgery, Key Laboratory of Developmental Genes and Human Disease, School of Life Science and Technology, Zhongda Hospital, Southeast University, Nanjing, China
[3] Institute for Regenerative Medicine, State Key Laboratory of Cardiology and Medical Innovation Center, Shanghai Key Laboratory of Signaling and Disease Research, Frontier Science Center for Stem Cell Research, School of Life Sciences and Technology, Shanghai East Hospital, Tongji University, Shanghai, China
[6] State Key Laboratory of Digital Medical Engineering, School of Biological Science & Medical Engineering, Southeast University, Nanjing, China
Full list of author information is available at the end of the article

## Abstract

Obtaining single-cell spatial information remains a challenge in spatial transcriptomics. Here we develop SEU-TCA, a method that leverages transfer component analysis to improve single-cell spatial mapping accuracy. Application to multiple single-cell and spatial transcriptomic datasets shows superior performance in spatial deconvolution and cell mapping. Using SEU-TCA, we explore spatial gene expression and regulon activity during mouse gastrulation and identify anterior second heart field progenitors regulated by *Irx1*. Functional experiments reveal that *Irx1* deletion disrupts anterior second heart field development and causes ventricular septal defects, underscoring SEU-TCA's potential for advancing developmental biology research.

**Keywords:** Single-cell RNA sequencing, Spatial transcriptomic, Cardiac progenitors, Regulatory factors

## Background

Understanding the precise spatial positions of individual cells with transcriptomic signatures during early developmental stages is instrumental in bridging cellular functions and their spatial contributions with developmental processes. Recently, numerous single-cell transcriptomic atlases [1, 2] and spatial transcriptomic (ST) maps [3–5] have been independently reported to delve into the early developmental processes. Nevertheless, a key limitation of single-cell RNA-sequencing (scRNA-seq) analysis lies in its requirement for tissue dissociation, which inevitably leads to the loss of spatial position information. In contrast, current ST technologies typically capture in situ gene expression within spots containing multiple cells, inherently precluding the achievement of single-cell resolution. Therefore, there is an urgent need for computational methodologies that precisely predict the associations between scRNA-seq profiled "cells" and spatially resolved "spots" from ST data.

To date, numerous statistical approaches have emerged to integrate scRNA-seq datasets with ST data. These methodologies can be categorized into two primary groups: deconvolution methods and mapping methods, distinguished by their integration strategies. Deconvolution methods primarily disentangle the mixture of cells within each spatial spot, leveraging a reference scRNA-seq dataset [6–14]. Examples include cell2location [9] and CARD [10], which specialize in estimating the proportions of various cell types or states within each spot. Nevertheless, the precise spatial positioning of cells within the scRNA-seq dataset remains unaddressed by these methods. Conversely, mapping methods employ reference ST data to infer and assign spatial position information to individual cells within the scRNA-seq dataset [15–20]. Widely used methods in this category, including Tangram [15], SpaGE [19], and Seurat [18], construct mathematical models to integrate scRNA-seq dataset with ST data. Tangram, a deep-learning framework, correlates single-cell gene expression profiles with ST data, enabling accurate spatial mapping of scRNA-seq data from a reference ST dataset [15]. SpaGE aligns ST and scRNA-seq data through domain adaptation using PRECISE [19]. Seurat, while primarily focusing on batch correction among scRNA-seq datasets from different experiments, employs anchoring and transfer learning techniques to project single-cell data onto spatial coordinates [18]. Although significant progress has been made, these existing methods are still limited in their ability to simultaneously minimize distributional disparities between single-cell and spatial data, extract spatially relevant feature representations, and identify gene regulatory networks at the individual cell level within their respective spatial locations.

There is an urgent need to precisely map scRNA-seq single-cells to ST spots, as both spatial information and single-cell resolution are indispensable for elucidating the highly dynamic cellular interactions and developmental events. For example, the gastrula stage, in particular, represents a critical juncture in embryonic development where the three germ layers—endoderm, mesoderm, and ectoderm—are established, setting the foundation for subsequent organogenesis. However, currently, the finest resolution of gastrula ST data is at the scale of 20–40 cells, as demonstrated by studies such as Geo-seq, which cover embryonic stages E5.5 to E7.5 [4, 5]. This resolution is insufficient to reconstruct the intricate cell type distribution observed during this pivotal developmental period. On the other hand, while the single-cell transcriptomic atlas of mouse early gastrulation (scRNA-seq, E6.5-E8.5) has been systematically annotated [1], it lacks the crucial spatial context to fully understand cellular dynamics within the gastrula.

Here we introduced Spatial Expression Utility—Transfer Component Analysis (SEU-TCA), an integration approach leveraging TCA [21] to extract shared features in a shared latent space of scRNA-seq and ST data. By applying SEU-TCA to four distinct biological systems—mouse gastrulation, human heart, mouse olfactory bulb, and pancreatic ductal adenocarcinoma—we demonstrated its superior performance over existing methods in deconvolving the cellular composition of ST spots and predicting spatial locations for single cells from scRNA-seq data. To further verify the practical utility of SEU-TCA, we conducted an in-depth exploration of the results obtained from the E7.5 gastrulation ST maps and scRNA-seq datasets, extending our analysis to infer the spatial distribution of regulon activities for mesoderm. Building upon this analysis, we further dissected cardiac development, the earliest organogenesis program initiated during

mesodermal development, at both the single-cell and spatial levels. We identified a series of spatially-specific transcription factors (TFs) and found that IRX1 specifically regulates the development of the anterior second heart field (aSHF). Genetic lineage tracing and gene knockout experiments revealed that Irx1-positive progenitors contributed to the aSHF lineage and its derivatives, and the deletion of *Irx1* led to ventricular septal defects (VSDs) in mice. Taken together, these results demonstrate that SEU-TCA is capable of precisely deconvolving the spatial organization and dynamics of cells, providing a robust framework for integrating single-cell and spatial transcriptomics to study developmental processes.

## Results

### SEU-TCA: a TCA-based method for spatial mapping

To simultaneously decipher the spatial heterogeneity and cellular organization within complex tissues, we developed SEU-TCA to establish virtual connections between single cells and spatially resolved spots by applying the TCA approach [21]. The primary motivation of SEU-TCA is to identify the optimal nonlinear transformation (ϕ) that maps both reference data ($X_R$, ST) and query ($X_Q$, scRNA-seq) data into a shared latent space, where the Maximum Mean Discrepancy (MMD) between the latent representations of the reference ($TC_R$) and query ($TC_Q$) is minimal (Fig. 1; Methods). Pearson correlation coefficient (PCC) between $TC_R$ and $TC_Q$ is calculated to evaluate the spot-cell similarity. To extract meaningful insights, SEU-TCA can be further extended to incorporate downstream analysis focusing on three aspects: (1) spot deconvolution, aimed at resolving cell heterogeneity of spots; (2) inferring the spatial location of target cells to explore spatial heterogeneity of single cells; and (3) identifying spatial regulon to construct spatially informed gene regulatory networks at single-cell resolution.

### SEU-TCA demonstrates superior performance to the existing methods

We systematically evaluated SEU-TCA's performance across four key dimensions: accuracy in simulation data, robustness to parameter variations, computational efficiency, and generalizability across diverse datasets.

### Accuracy validation using simulation data

To generate benchmark data for accuracy evaluation, we applied a grid-based approach on a single-cell spatial transcriptomics dataset of human heart [22] (Fig. 2A; Methods). Specifically, the average expression of all cardiomyocytes (CMs) within each grid was used as the pseudo-bulk expression for each spot, while the dominant cell type within each grid was taken as the ground truth for each spot. We applied SEU-TCA to the pseudo-spot-level data and the corresponding single-cell CM data from the human heart dataset to predict cell types and spot composition. The performance was evaluated using multiple metrics, including PCC (to assess the consistency of expression levels between spot-cell pairs), Accuracy (ACC) and F1 score (to provide a balanced evaluation of prediction performance), Sensitivity and Specificity (to measure the method's ability to correctly identify true positive and true negative cell types), and Adjusted Rand Index (ARI, to evaluate the similarity between predicted clustering and ground truth).

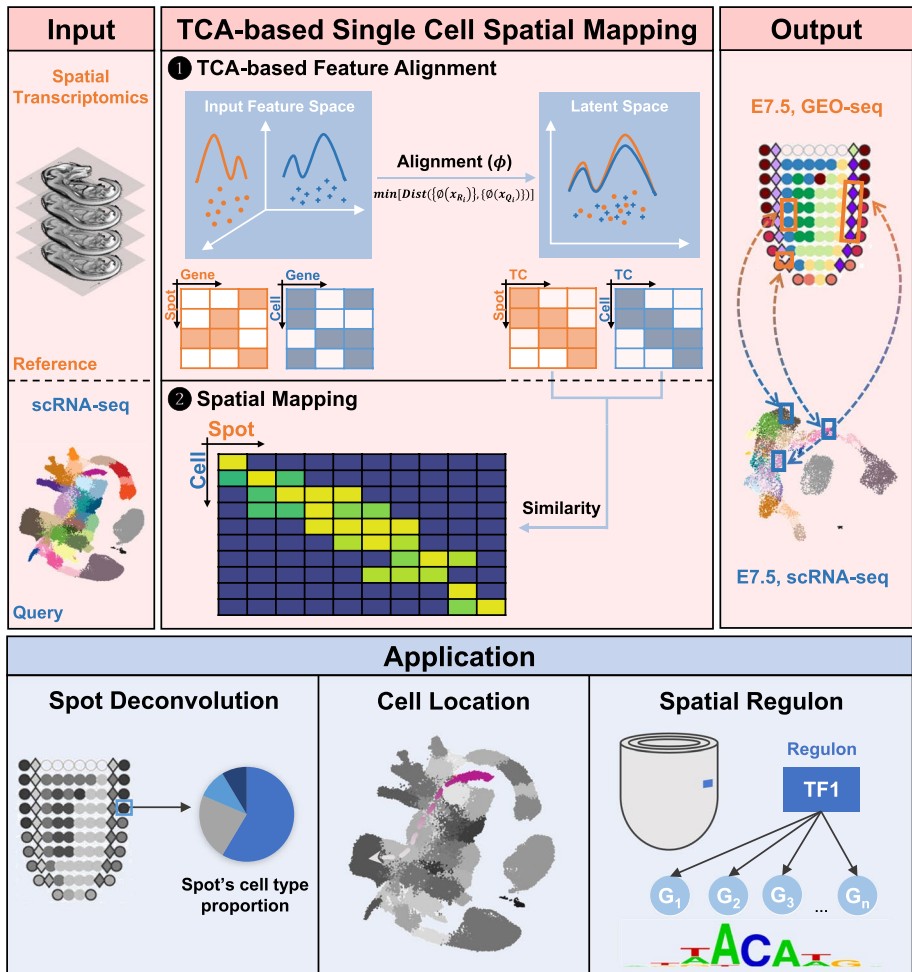

**Fig. 1** TCA-based method for single-cell spatial mapping. Overview of methodology (pink area) and downstream application (blue area) of SEU-TCA. Methodology: Step 1) Given spatial transcriptome as reference ($X_R$) and single-cell transcriptome as query ($X_Q$), SEU-TCA applies TCA analysis and finds common latent representation ($TC_R$ and $TC_Q$) of the two datasets; Step 2) PCC between $TC_R$ and $TC_Q$ is then computed to evaluate the similarity between spots in the spatial transcriptome and cells in the single-cell transcriptome, enabling spatial mapping. The output of SEU-TCA is shown using the example of E7.5 mouse embryo's Geo-seq (spatial transcriptome, each spot contains 5–40 cells) and scRNA-seq. Downstream applications: 1) resolving cell heterogeneity of spots; 2) inferring spatial information of single-cells; and 3) identifying spatially variable genes

We first compared SEU-TCA with two single-cell mapping methods (Tangram [15] and SpaGE [19]) and four deconvolution methods (CARD [10], cell2location [9], STRIDE [11], and CIBERSORTx [23]) on this human heart dataset. Here, the predictions by SEU-TCA were closely aligned with the ground truth, accurately capturing the spatial organization and boundaries of cell types (Fig. 2B). This alignment demonstrated the superior overall performance of SEU-TCA across the evaluation metrics (Fig. 2B, C; Additional file 1: Fig. S1). Among all compared methods, SEU-TCA shows the highest ARI value (0.64), followed by SpaGE (0.52), Tangram (0.49), cell2location (0.43), STRIDE (0.40), CARD (0.40), and CIBERSORTx (0.09) (Fig. 2B). We also observed that, except for CIBERSORTx, all methods were able to successfully infer dominant cell types with clear distinctions between the left and right atria (LA and RA). However, for the left

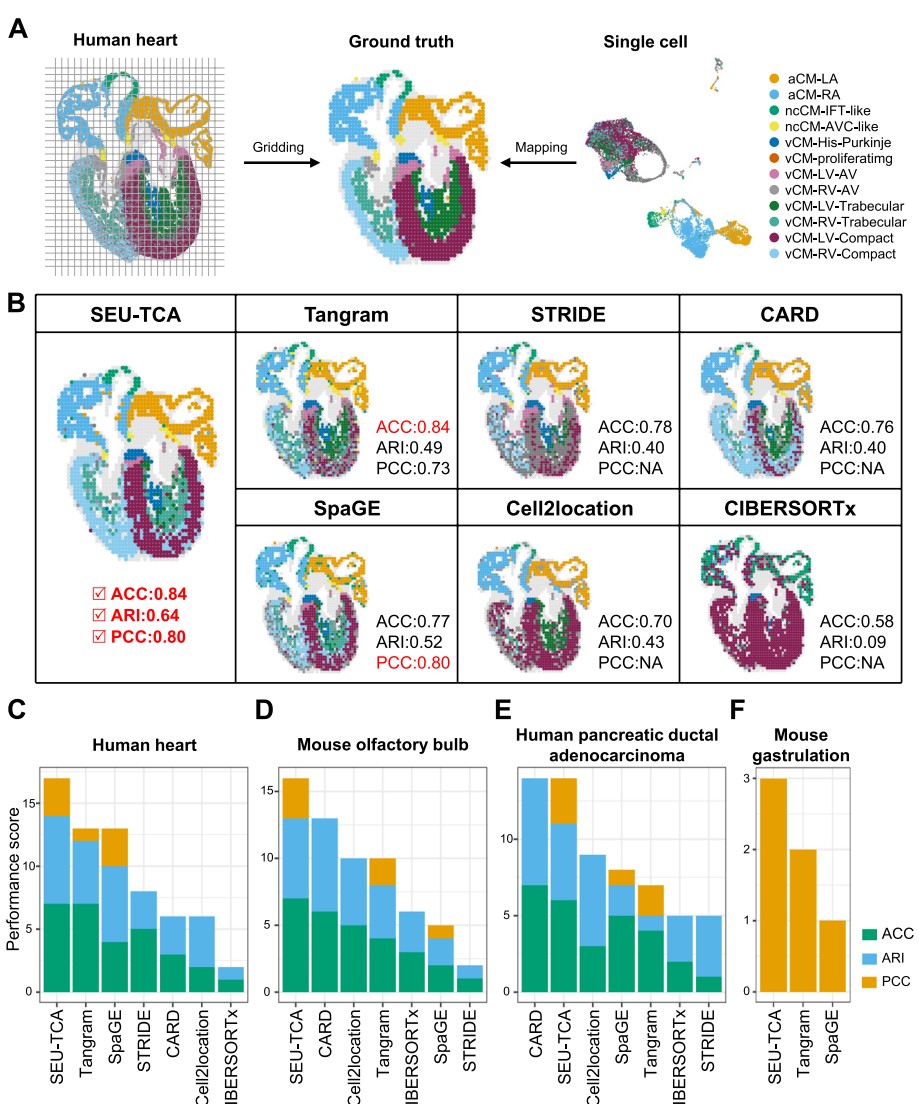

**Fig. 2** SEU-TCA demonstrates superior performance to existing methods. **A** The procedure for generating pseudo-bulk spots from the human heart dataset using a grid-based approach (Methods). The left panel displays the actual MERFISH cells in a frontal section of a developing human heart. The middle panel shows the pseudo-bulk spots created after gridding, with the cell type having the highest proportion within each spot taken as the ground truth. The right panel presents the single-cell data matched with the ST data, focusing exclusively on CMs for this analysis. The single-cell data were downsampled to 10,000 cells to create a manageable dataset for the subsequent analysis, ensuring sufficient coverage while maintaining computational efficiency. **B** Dominant cell types inferred using seven methods are shown, with corresponding PCC, ACC, and ARI metrics calculated by comparing predictions to the ground truth. PCC values are marked as NA for deconvolution methods, as these methods cannot obtain corresponding spot-cell pairs, making it impossible to compute the correlation of expression levels between spots from ST and cells from SC data. **C**- **F** Performance Summary: The evaluation across the four datasets was conducted by assigning scores to each method based on their rankings in three key metrics: PCC, ACC, and ARI. The scoring system assigned 7 points to the top-ranked method, 6 points to the second, and so forth, decreasing by 1 point per rank. These scores were aggregated to provide a comprehensive performance summary for each method across the human heart dataset (**C**), the mouse olfactory bulb dataset (**D**), the pancreatic ductal adenocarcinoma dataset (**E**), and the mouse gastrulation dataset (**F**)

and right ventricles (LV and RV), SEU-TCA achieved more precise predictions, particularly for ventricular cardiomyocyte (vCM)-RV-Trabecular and vCM-LV-Trabecular. Additionally, careful examination of the cell type composition for each spot further confirmed the accuracy of SEU-TCA deconvolution (Additional file 1: Fig. S1B). Similarly, SEU-TCA achieved strong performance (median PCC $=0.80$), matching the performance of SpaGE (median PCC $=0.80$), outperforming Tangram (median PCC $=0.73$) by 10% (Fig. 2B; Additional file 1: Fig. S1 C). Other methods did not report PCC values because they are designed as spot deconvolution algorithms, providing the cell type composition for each spot rather than the relationships between spots and cells. These results highlight the great capability of SEU-TCA to reconstruct the ground truth cell type distribution with high precision and clearly distinguish critical anatomical regions.

### Robustness to parameter variations

SEU-TCA offers three kernel types (Primal, Linear, RBF) for mapping data into the latent space, along with parameters such as the RBF kernel bandwidth ($\gamma$) to flexibly adapt to diverse data distributions. We also conducted extensive evaluations of kernel performance and parameter robustness using the human heart dataset. For the parameter alignment dimensions, we chose values ranging from 10 to 100 in increment of 10 under three kernel types (Primal, Linear, and RBF) and assessed their performance using ACC, F1 score, Sensitivity, and Specificity (Additional file 1: Fig. S2 A). The results demonstrated that dimensions above 30 had minimal impact on the performance, and the Primal kernel consistently outperformed both Linear and RBF kernels in this dataset. Additionally, we performed robustness check of the bandwidth parameter ($\gamma$) for the RBF kernel. Grid search analyses show minimal changes in performance with different $\gamma$ values (Additional file 1: Fig. S2B). The abovementioned results suggest that SEU-TCA is robust to parameter variations across a wide range of settings.

### Computational efficiency

The computational efficiency of SEU-TCA was evaluated on the human heart dataset. SEU-TCA achieved faster runtime compared to several other methods, substantially reducing the time required for analysis (Additional file 1: Fig. S3 A). Furthermore, SEU-TCA consumed ~1 Gb memory for this task with 10,000 single cells, demonstrating its efficiency in resource utilization (Additional file 1: Fig. S3B). These findings highlight the practical advantages of SEU-TCA and established it as a scalable solution for fine-map reconstruction in much larger datasets.

### Generalizability across biological systems

To demonstrate SEU-TCA's applicability on real ST data, we first applied it to the mouse olfactory bulb data [24], which clearly defined four main anatomic layers: the granule cell layer (GCL), the mitral cell layer (MCL), the glomerular layer (GL), and the nerve layer (ONL) (Additional file 1: Fig. S4 A). Single-cell data from 10x Chromium on the same tissue [25] was utilized for the analysis. The results inferred by SEU-TCA clearly revealed the expected layered structure of the mouse olfactory bulb, as shown in the visualizations of the dominant layers (Additional file 1: Fig. S4B). In comparison, although CARD and cell2location achieved relatively accurate layering, they exhibited less clarity

in defining the boundary between the GL and ONL (Additional file 1: Fig. S4 C). Tangram produced blurred transitions between layers, and CIBERSORTx struggled to capture the distinct structural organization. Notably, SpaGE and STRIDE failed to differentiate the boundaries between most regions. Additionally, the evaluation metrics further confirmed the superior performance of SEU-TCA over other methods (Additional file 1: Fig. S4D-F). For example, it achieved the highest PCC, reflecting its strong ability to maintain consistent gene expression levels between spots and single cells. SEU-TCA also outperformed all other methods in terms of ACC, F1 score, and Sensitivity, while achieving comparable ARI and Specificity to CARD, demonstrating its remarkable stability and reliability. Overall, SEU-TCA provides the notably precise reconstruction of the mouse olfactory bulb's layered architecture, particularly in distinguishing the ONL from adjacent layers.

To further explore its applicability in complex human pathological tissues, we then evaluated the performance of SEU-TCA on the pancreatic ductal adenocarcinoma dataset obtained from ST technology and its matched single-cell dataset generated via the inDrop platform [26]. The pancreatic ductal adenocarcinoma dataset consists of three regions with clear boundaries—ductal cells, acinar cells, and cancer—along with a stromal region, all labeled based on histological annotations (Additional file 1: Fig. S5 A). As expected, SEU-TCA, as well as CARD and STRIDE, was able to precisely delineate the complex architecture of the pancreatic ductal adenocarcinoma (Additional file 1: Fig. S5B-C). In contrast, Tangram and cell2location produced ambiguous transitions between cancer and acinar cells, making it difficult to discern the exact cellular composition. CIBERSORTx failed to identify specific cell types within the pancreatic ductal adenocarcinoma, while SpaGE had limited capability of differentiating the distinct regions. To quantify the performance of SEU-TCA over other methods, we calculated six key metrics: F1 score, ACC, ARI, Sensitivity, Specificity, and PCC for these methods on this pancreatic ductal adenocarcinoma data (Additional file 1: Fig. S5D-F). In terms of the F1 score (Additional file 1: Fig. S5D), SEU-TCA once again outperformed all competing approaches, underscoring its proficiency in providing highly accurate predictions. Furthermore, SEU-TCA ranked among the top three methods in terms of ACC, ARI, Sensitivity, and Specificity, reflecting its robustness and reliability in precisely identifying distinct cell types of the pancreatic ductal adenocarcinoma. For the PCC (Additional file 1: Fig. S5E), SEU-TCA achieved the highest value among three methods, demonstrating its unique ability to capture the intricate patterns of cell type distribution. In summary, the comprehensive evaluation of SEU-TCA across multiple metrics confirms its robust and reliable performance in analyzing the pancreatic ductal adenocarcinoma data, consistently surpassing or matching the performance of other existing methods.

To overcome the inadequacy of single-cell resolution gastrula ST data and the lack of spatial information in the gastrula scRNA-seq atlas, we next applied SEU-TCA to analyze Geo-seq and scRNA-seq data for E7.5 (0B-EB stage) mouse embryos, a critical developmental period characterized by gastrulation, during which the embryonic germ layers are formed, and the body axes are established [27]. SEU-TCA achieved high PCC values (mesoderm $0.84 \pm 0.13$; endoderm $0.85 \pm 0.12$; ectoderm $0.78 \pm 0.13$) and accurately recovered the original expression patterns (Additional file 1: Fig. S6). In comparisons with other representative single-cell mapping algorithms, such as Tangram [15]

and SpaGE [19], SEU-TCA yielded more accurate estimation of the expression levels and the cell type proportions across spatial locations, underscoring the importance of feature alignment for effective data integration. Moreover, SpaGE, which also aligns features, outperformed Tangram but remained less accurate than SEU-TCA (Additional file 1: Fig. S7 A). In contrast, CIBERSORTx—dependent on predefined cell-type specificity—exhibited discrepancies in both spatial localization and cell type proportion estimation. (Additional file 1: Fig. S7B-D). Taken together, SEU-TCA demonstrates superior accuracy in integrating spatial and single-cell transcriptomics for the multiple datasets from distinct biological systems, consistently ranking among the top methods in terms of ACC, PCC, and ARI (Fig. 2C–F).

Overall, these results spotlight SEU-TCA's outstanding accuracy, robustness, efficiency, and generalizability, as evidenced by its consistent excellence across various metrics. Through comprehensive evaluations on the human heart, the mouse olfactory bulb, the pancreatic ductal adenocarcinoma, and the mouse gastrulation datasets, SEU-TCA has proven its reliability, scalability, and practical benefits in reconstructing complex spatial and pathological tissue architectures.

### SEU-TCA decodes spatial cell heterogeneity

SEU-TCA is a method capable of inferring spatial context for the single-cell transcriptomic landscape. To demonstrate its utility, we applied SEU-TCA to predict cellular locations of E7.5 mesodermal scRNA-seq using Geo-seq data as reference. Geo-seq spots were spatially divided into four zones, namely Proximal-Anterior zone (P-A), Proximal-Posterior zone (P-P), Distal-Anterior zone (D-A), and Distal-Posterior zone (D-P). Single cells with the same predicted zone identity were clustered together in the transcriptomic space, as evidenced by the space-independent UMAP (Fig. 3A and Additional file 1: Fig. S8 A). Moreover, the associated genes of the top two TCs exhibited regionalized expression patterns just along the P-D and A-P axes, respectively (Additional file 1: Fig. S8B).

(See figure on next page.)

**Fig. 3** SEU-TCA decodes mesodermal cellular heterogeneity. **A** UMAP layout for the E7.5 mesodermal cells from Pijuan-Sala et al. is colored by four quadrants constructed by A-P and P-D axes (upper), predicted by SEU-TCA, and cell type (lower). The dashed lines represent cell clusters along the developmental axis. Embryonic axes: anterior–posterior, A-P; proximal–distal, P-D. NM, nascent mesoderm. MM, mixed mesoderm. IM, intermediate mesoderm. PhM, pharyngeal mesoderm. Mch, mesenchyme. PaM, paraxial mesoderm. SM, somitic mesoderm. EM, extraembryonic mesoderm. **B** Corn plots (top row, Geo-seq) and UMAP (bottom row, scRNA-seq) showing the mesodermal spatial expression patterns and single-cell expression levels of *Krt8*,*Otx2*, *Cdx4* and *Hes7* at E7.5, which marks four quadrants constructed by A-P and P-D axes. MA, anterior mesoderm; MP, posterior mesoderm. **C** Fraction of cell type predicted from E7.5 mesodermal cells for each Geo-seq spot, e.g. 10MA representing the 10th slice of anterior mesoderm. **D** DEGs between 6MA and 8MA for E7.5 scRNA-seq data are expressed at 7MA. **E** UMAP layout for 7MA cells inferred from E7.5 mesodermal Geo-seq is colored by cell type. The dashed line circles all 7MA cells in E7.5 mesodermal scRNA-seq. **F** Corn plots showing the spatial pattern of expression of 7MA markers and UMAP showing the single-cell resolution pattern of expression of 7MA markers at E7.5. **G** Fraction of inferred location for each E7.5 mesodermal cell type. Each cell type is arranged from top to bottom according to its proximity to the anterior spots and distance to the posterior spots. The varying shades of color at both ends of the bidirectional arrow represent the proximity to the anterior (red) and the posterior ends (blue). **H** Corn plot showing the spatial pattern of inferred contributions of NM at E7.5. **I** UMAP showing the results of NM cells re-clustered based on the spatial feature genes. The dashed line circles all NM cells in E7.5 mesodermal scRNA-seq. Heatmap visualizing the abundance of subclusters of NM in each spot in Geo-seq data. Rows represent subclusters, columns represent spatial locations. **J** Dot plot showing the key markers of subclusters of NM

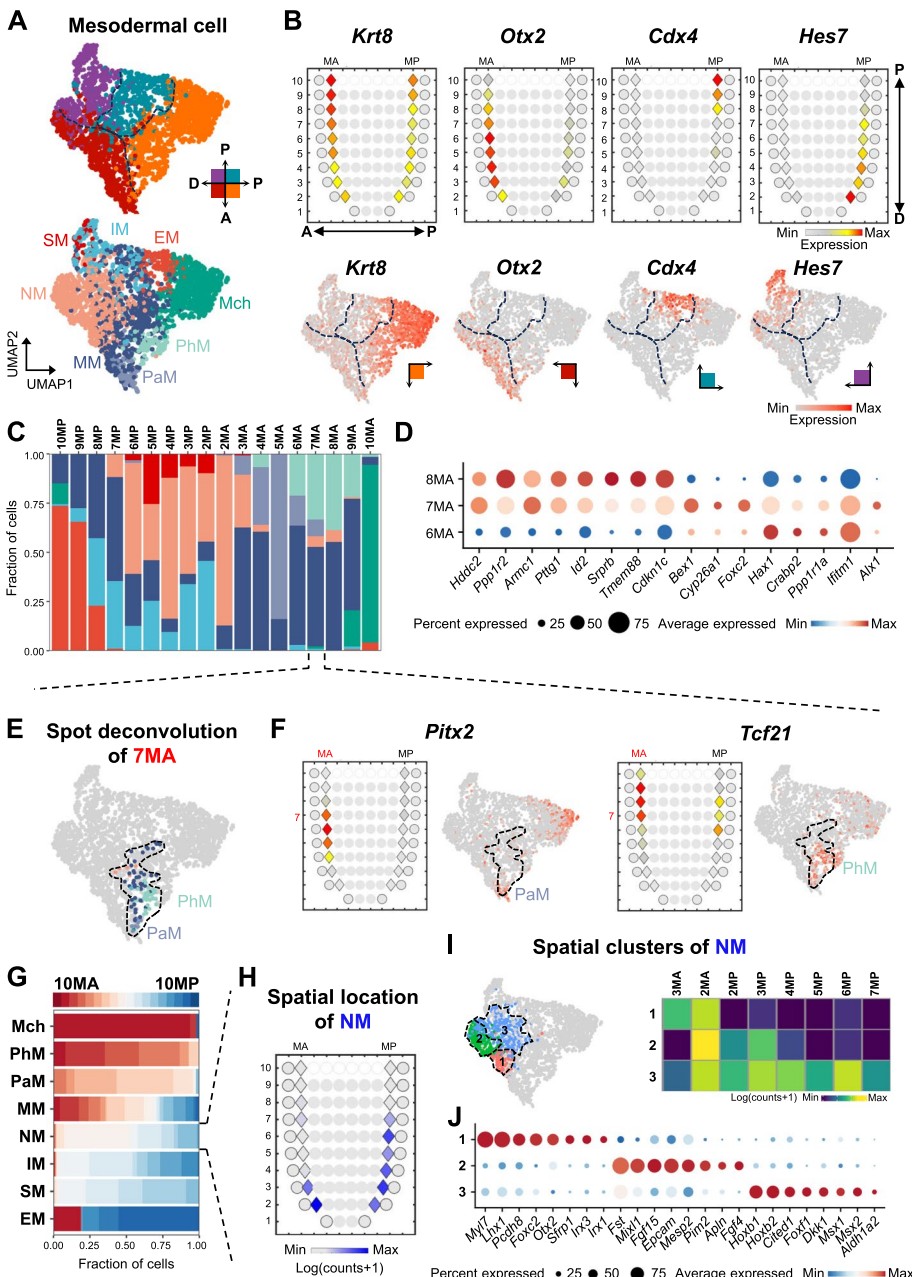

**Fig. 3** (See legend on previous page.)

These results strongly support the association between spatial and molecular variations of the E7.5 mesodermal cellular landscape.

We further identified markers for each spatial cluster, which exhibited highly regionalized and variable expression patterns in Geo-seq and scRNA-seq data, respectively (Fig. 3B). In total, we identified 123 genes with high region-specific expression, many of which have not been previously reported, while others have been reported as region-specific or driver genes underling embryonic pattern formation (Additional file 2: Table S1). For example, the D-A marker gene *Krt8* is expressed in the extraembryonic mesoderm and plays a crucial role in the establishment of the A-P axis [28, 29]; *Otx2*, a

V-A marker, is expressed in the anterior mesoderm (AM) [30–32]; *Cdx4*, a D-P marker, is expressed posteriorly and functions during embryonic axial formation [33, 34]; and *Hes7*, a V-P marker, is dynamically expressed in the pre-somitic mesoderm (PSM) during somitogenesis [35, 36].

By mapping single cells to ST spots representing mixtures of multiple cells, we were able to quantify the cell type composition of each spot. Along the caudal-rostral axis, following the order of 10MP-2MP-2MA-10MA (from the posterior-most end of the mesoderm to its anterior-most end), the distribution of cell types exhibited a discernible and continuous pattern. The transition of major cell types, excluding the less mature nascent mesoderm (NM) and mixed mesoderm (MM), formed the following order: extraembryonic mesoderm (EM)—intermediate mesoderm (IM)—somitic mesoderm (SM)—paraxial mesoderm (PaM)—pharyngeal mesoderm (PhM)—mesenchyme (Mch) (Fig. 3C). At most spots, the proportion of the major cell type was less than 50%, underscoring the necessity of deconvolution. In addition, we found that complex cell type compositions were more likely to occur at the junctions of multiple tissues. An example was the 7MA location, which contained 6 cell types, including relatively mature and predominant cell types such as PaM and PhM (Fig. 3C, E). Additionally, we found that 7MA simultaneously expressed marker genes associated with both 6MA and 8MA (Fig. 3D). In the 2-day-old chick embryo, PhM is primarily divided into two overlapping subdomains: the mesenchymal paraxial mesoderm (mPaM) and the medial splanchnic mesoderm (mScM) [37]. mPaM and mScM specifically express *Pitx2* and *Tcf21*, respectively [37–40]. Interestingly, 7MA expresses both *Pitx2* and *Tcf21*, whereas in scRNA-seq data, these two genes are specifically expressed in PaM and PhM cells, respectively (Fig. 3F). These findings suggest that the co-expression of PaM and PhM marker genes at 7MA results from the spatial overlap of the two cell types rather than the presence of intermediate cell states. Collectively, SEU-TCA provides valuable insights into cell type distribution, transitions, and spatial overlap.

### SEU-TCA facilitates the identification of subtypes

We further systematically analyzed the spatial distribution of different cell types, the predicted positions of which were consistent with those reported in the literature [41] (Fig. 3G; Additional file 1: Fig. S9). Most cell types exhibited a short-strip-like distribution, while others, such as IM and NM, displayed a long-strip-like pattern. NM, the first mesodermal cell type originating from the primitive streak, was located across the A-P axis and near the distal side [42] (Fig. 3H). To further investigate the spatial heterogeneity of NM, we re-clustered NM cells using spatial feature genes, identifying three distinct spatial distribution clusters (Fig. 3I). We found that the cells in cluster 1 were exclusively localized at the anterior side (3MA-2MA) and specifically expressed *Lhx1*, *Otx2*, and *Sfrp1* (Fig. 3J). The expression of *Lhx1* in the epiblast marks the anterior mesendoderm. Together with *Otx2* and *Sfrp1*, *Lhx1* plays a crucial role in regulating its development [43]. In contrast, cells in cluster 2 were distributed along the A-P axis (2MA-2MP-3MP) and characterized by the expression of *Fst*, *Epcam*, *Pim2*, and *Apln*, which were expressed in neuromesodermal progenitors (NMP)-fated cells [44]. However, the spatial distribution of cluster 3 was broader (2MA-2MP-3MP-4MP-5MP-6MP-7MP), suggesting the possibility of multiple distinct developmental fates. Within this cluster, *Hoxb1*,

*Aldh1a2*, and *Foxf1* mark the posterior second heart field (pSHF) cells, while *Msx1/2* and *Cited1* mark the first heart field (FHF) cells, indicating that cluster 3 is likely a common progenitor for both FHF and pSHF [45]. These findings confirm that incorporating spatial information into single-cell data can further uncover spatially specific cell populations.

### SEU-TCA enables construction of the spatial regulon map

To investigate the gene regulatory network underlying the spatially patterned mesodermal cell types, we set out to estimate the regulon activity at each ST spot. We first utilized the single-cell regulatory network inference and clustering (SCENIC) pipeline [46] to assess the regulon activity score (RAS) for individual mesodermal single-cells. We then generated a regulon activity heatmap, where single-cells were ordered by spatial locations and regulons were ordered using hierarchical clustering (Additional file 1: Fig. S10 A). Our analysis revealed distinct clusters of regulons characteristic of specific spatial spots or regions of consecutive spots. For example, Lhx1(+) represented a cluster of regulons with high RAS at 2MA and 3MA (Additional file 1: Fig. S10 A, B), consistent with its functions in governing anterior mesendoderm development [43, 47]. Another regulon cluster, represented by Hoxa10(+), showed specific and strong activity in EM located at 10MA and 10MP (Additional file 1: Fig. S10 A, B), consistent with its high enrichment in accessible chromatin of EM [48]. In contrast, a regulon cluster consisting of Gm10093(+) was specifically depleted in EM and exhibited high activity across the embryonic portion of mesoderm (Additional file 1: Fig. S10 A). Within the embryonic part, a cluster including Zfp467(+) was highly active in the distal part spanning from 2MA to 5MP (Additional file 1: Fig. S10 A). These findings suggest that mesodermal spatial pattern could be composed of transcriptional regulation along both embryonic-extraembryonic and rostral-caudal axes.

We further constructed a spatial regulon activity map for the mesoderm by averaging the RAS of single cells mapped to each spot and identified significantly enriched regulons in each spot (Fig. 4A; Additional file 3: Table S2). This allowed us to identify key gene regulatory networks functioning in a spatial location-dependent manner. Based on this spatial regulon map, we observed specific enrichment of the *Hox* family members at 10MA and 10MP (Fig. 4A), consistent with previous reports indicating that the *Hox* genes exhibit a temporally co-linear expression pattern during the differentiation of the posterior primitive streak, ultimately leading to EM formation [49]. In addition, we discovered that the *Fox* family members, which are crucial for embryonic development [50], exhibited highly specific regulon activity at 5MA and 6MA (Fig. 4A). For instance, *Foxa2* has been reported to be expressed on the anterior side at E7.75 and is necessary for ventricular cell generation during cardiac development [51].

After ranking regulons by the regulon specific score (RSS) for each spot, we identified candidate TFs driving location-specific cellular specification (Additional file 1: Fig. S11). For example, the 3MP spot, primarily composed of SM and the undifferentiated NM cells (Fig. 3C), exhibited high regulon activities of *Sp5* and *Tbx6* (Fig. 4B). The corresponding RAS and the TF expression was consistent at both spatial and transcriptomic levels (Fig. 4C–F). Previous studies have suggested that the expression of *Sp5* is correlated with NM formation and is dynamically and restrictively expressed during

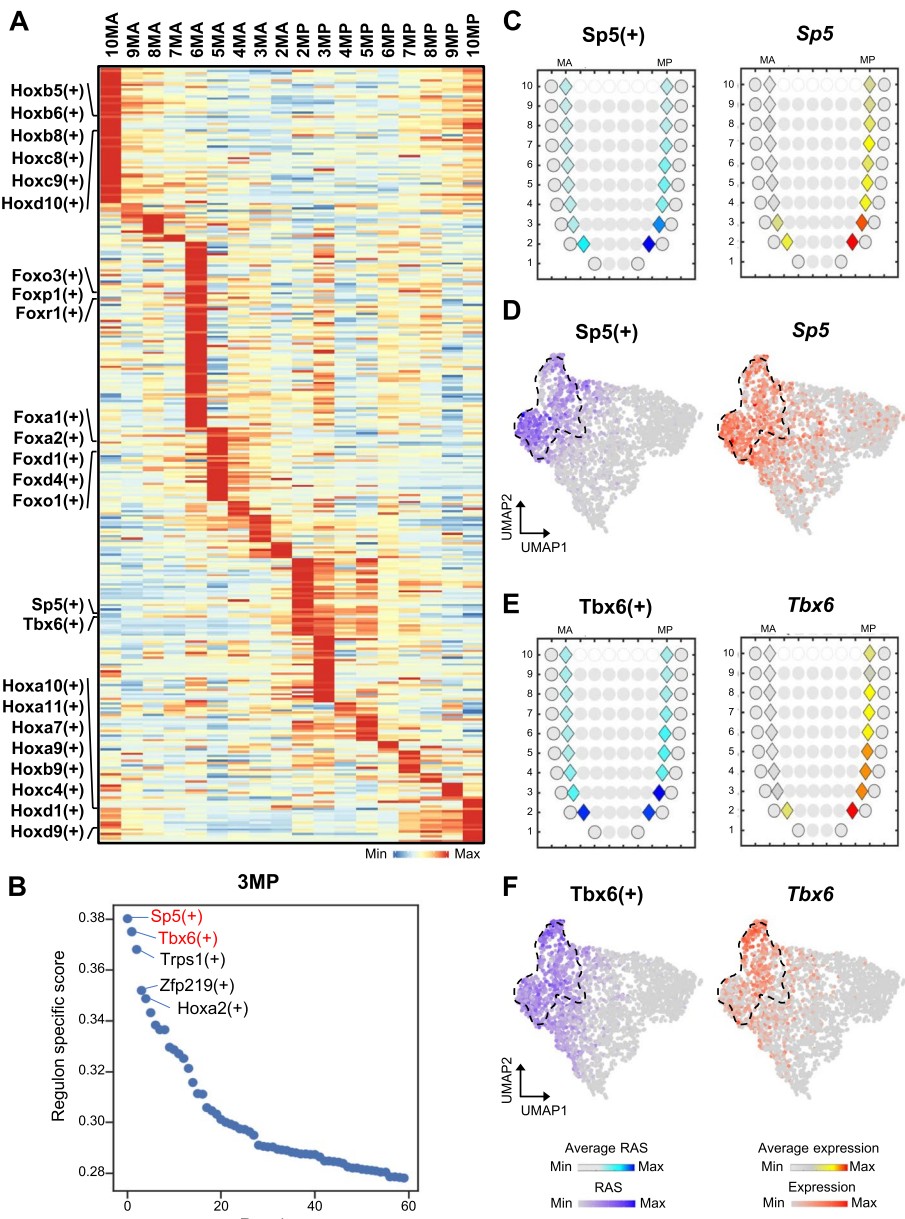

**Fig. 4** SEU-TCA enables construction of the spatial regulon maps. **A** Heatmap illustrates the spatial regulon map, with rows representing regulons and columns representing spatial locations. The redder the color, the higher the average RAS. **B** The top five regulons in 3MP are labeled on the plot. The y-axis displays the specificity score. **C** Corn plots showing the spatial pattern of regulon activity (left) and expression (right) of *Sp5* in E7.5 mesodermal Geo-seq. **D** UMAP showing the single-cell resolution pattern of regulon activity (left) and expression (right) of *Sp5* in E7.5 mesodermal scRNA-seq. **E** Corn plots showing the spatial pattern of regulon activity (left) and expression (right) of *Tbx6* in E7.5 mesodermal Geo-seq. **F** UMAP showing the single-cell resolution pattern of regulon activity (left) and expression (right) of *Tbx6* in E7.5 mesodermal scRNA-seq

somitogenesis [52, 53]. Furthermore, it has been shown that *Tbx6* is highly expressed in the PSM and that loss of *Tbx6* leads to severe defects in somitic development [44, 54, 55]. The spatial regulon map also revealed a large number of spot-specific regulons and

TFs that had previously been unrecognized, which might be potential candidates for further functional validation.

### SEU-TCA predicts cardiac progenitor cell locations

Cardiac development is a complex process involving multiple lineages [56], yet where and when progenitor cells segregate remain unclear [57]. In our previous study, we employed Waddington-Optimal-Transport (WOT) analysis [58], a differentiation trajectory inference algorithm, to uncover the transcriptional trajectories and epigenetic determinants that specify early cardiac lineages from mesoderm [59]. To systematically trace the locations of cardiac progenitors across developmental stages, we combined SEU-TCA with WOT analysis, to explore the spatial organization of differentiation trajectories. Specifically, we first identified single cells of each cardiac lineage in the E8.5 mouse embryo, focusing on Nkx2-5-positive CMs and CM-progenitors, including Mab21l2-positive Mch [60] and Isl1-positive PhM [61] (Additional file 1: Fig. S12 A). These cells were further classified into nine distinct clusters (Fig. 5A–C). Our results suggest the existence of three major progenitor clusters at E8.5: the juxta-cardiac field (JCF), aSHF, and pSHF. The JCF is defined by the expression of *Mab21l2* [60] and *Hand1* [62], while the pSHF exhibits enriched expression of markers including *Osr1* [63] and *Nr2f2* [64], reflecting its posterior spatial identity and lineage commitment. The aSHF, on the other hand, is characterized by genes such as *Isl1* [61], *Tbx1* [65], *Fgf8* [66], and *Tcf21* [38], which collectively underscore their pivotal roles in second heart field progenitor specification and outflow tract (OFT) development (Fig. 5C; Additional file 4: Table S3). To visualize their developmental process, we performed WOT analyses on these progenitors and generated time-series tSNE maps (Fig. 5D; Additional file 1: Fig. S12B). Notably, we observed that cells of these lineages formed distinct clusters as early as E7.0. However, the lineage-specific markers did not initiate expression until E7.75 (Additional file 1: Fig. S13). Our findings suggest that progenitor cells from distinct spatial domains may begin to exhibit subtle transcriptomic differences, indicative of early lineage bias, even before the activation of canonical marker genes.

(See figure on next page.)

**Fig. 5** Spatial facilitates prediction of cardiac progenitor cell locations. **A** Time-series tSNE layouts showing the dynamic differentiation process of mesodermal lineage from E7.0 to E8.5. Epi, epiblast. PS, primitive streak. Ant.PS, anterior primitive streak. Haem, haematoendothelial progenitors. PGC, primordial germ cell. Def.end, definitive endoderm. RN, rostral neurectoderm. SE, surface ectoderm. C.Epi, caudal epiblast. AL, allantois. **B** Subcluster with spatial characteristics in E8.5 cardiac progenitors and CMs. The Mab21l2-positive Mch cells, Isl1-positive PM cells, and Nkx2-5-positive CMs are re-clustered into three key cardiac lineages. Among them, JCF mainly contributes to LV and AVC; aSHF mainly contributes to OFT and RV; pSHF mainly contributes to atria and SV. JCF, juxta-cardiac field. aSHF, anterior second heart field. pSHF, posterior second heart field. OFT, out flow tract. RV, right ventricle. LV, left ventricle. AVC, atrioventricular canal. SV, sinus venosus. **C** Dot plot showing the key markers of subclusters of cardiac progenitors and CMs. **D** Backward-tracing identifies development trajectory of JCF (blue), aSHF (yellow) and pSHF (green) during E7.0-E8.5 using the WOT analysis. E8.5 JCF, aSHF and pSHF were used as trajectory endpoints. At each time point, all cells are preliminarily screened based on a WOT score greater than 0.0001. The highest score among the three lineage contributions is selected for each cell. **E** The top corn plots showing the spatial pattern of inferred progenitors of JCF (left), aSHF (middle), and pSHF (right) at E7.5. In each corn plot, each diamond's color represents the weighted estimated cell type composition. The bottom corn plots showing the spatial pattern of expression of *Ahnak* (JCF progenitors), *Tbx1* (aSHF progenitors),and *Hoxb1* (pSHF progenitors) in E7.5 mesodermal Geo-seq

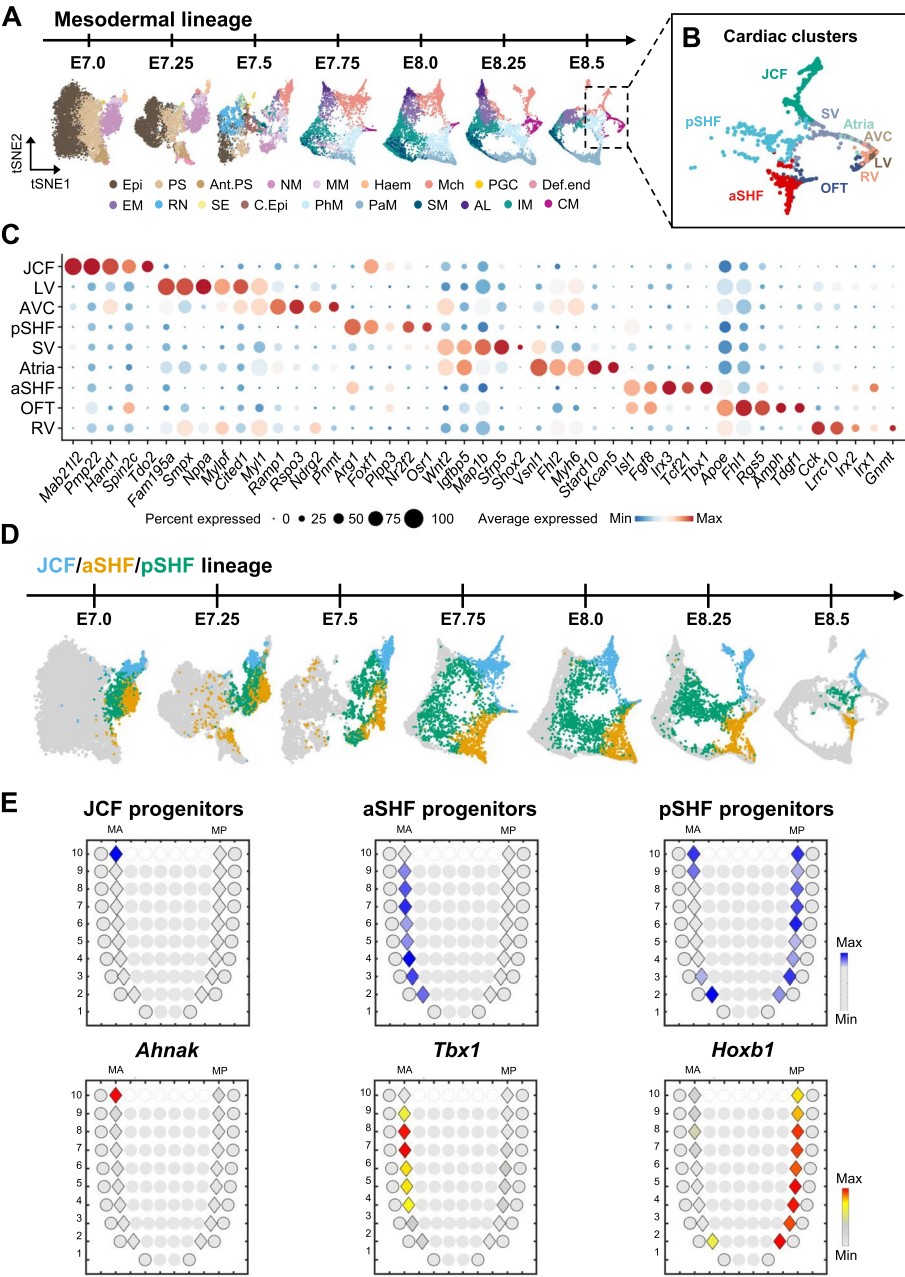

**Fig. 5** (See legend on previous page.)

To explore whether the cardiac lineages are also spatially segregated at early stages, we performed SEU-TCA analysis to map E7.5 cardiac progenitor cells onto E7.5 Geo-seq ST atlas (Fig. 5E). The mapping results revealed highly distinct spatial localization of different lineage progenitors. The JCF progenitors exhibited focused localization at 10MA, the anterior boundary between the forming cardiac crescent and extraembryonic tissue. In contrast, progenitors of aSHF and pSHF were spatially distributed across the anterior and posterior sides of the mesoderm, respectively. The predicted locations were supported by the expression patterns of the spatial genes including *Ahnak* for JCF progenitors [67, 68], *Tbx1* for aSHF progenitors [65], and *Hoxb1* for pSHF progenitors [69,

70] (Fig. 5E). The spatial predictions were also consistent with the expression patterns of TFs involved in lineage specification (Additional file 1: Fig. S14). For example, *Msx2* was specifically enriched in JCF (Additional file 1: Fig. S14 A), which has been identified in the gene regulatory network of JCF progenitors using epicardioids models [67]. *Foxc2* [71] and *Hoxb1* [69], which are early markers for aSHF and pSHF lineages, respectively, exhibited consistency in spatial pattern and expression pattern at aSHF and pSHF progenitors (Additional file 1: Fig. S14B, C). To date, this study represents the first systematic spatiotemporal tracing of early cardiac developmental trajectories.

### SEC-TCA reveals the potential function of *Irx1* in the aSHF lineage

Dysregulation or abnormalities in the development of the aSHF lineage have been associated with various congenital heart diseases (CHDs), including defects in the aorta, pulmonary artery, and ventricular septum [72, 73]. However, the subclusters and key regulatory factors of aSHF require further in-depth exploration. At E7.5, aSHF progenitors spatially spanned large areas of the anterior mesoderm (Fig. 5E), indicating the heterogeneity of aSHF. To spatially dissect the aSHF and identify functional regulators in aSHF development, we re-clustered the E7.5 aSHF progenitors based on the spatial feature genes derived from the ST atlas. This analysis resulted in three clusters, referred to as C1-C3 (Fig. 6A; Methods). After mapping the spatial locations of each cluster, we observed a consecutive distal-proximal order from C1 to C3, indicating a gradual transition of cell states along the P-D axis (Fig. 6B). Differentially expressed gene (DEG) analyses supported the distinction among the subclusters of aSHF (Fig. 6C). In the C1 subcluster, mesodermal TFs, such *Eomes* and *T*, were specifically expressed, suggesting they could be the aSHF progenitors just transitioning from mesoderm and lagging behind the rest of the subclusters, consistent with their posterior localization (Fig. 6B). *Foxc2* [71], reported as an aSHF-specific TF gene, was enriched in C2 (Fig. 6C).

Interestingly, the Iroquois homeobox (*Irx*) family members *Irx1*, *Irx3*, and *Irx5* were highly and specifically expressed in C2. Although the *Irx* family genes have been shown indispensable for heart development [74–76], their functions in early cardiac development remain unclear. Compared to *Irx3* and *Irx5*, the spatial pattern of expression and regulon activity of *Irx1* in the mesoderm exhibited higher spatial specificity, as suggested by Geo-seq data (Additional file 1: Fig. S15 A-C). RNAscope analyses also supported the specific expression of *Irx1* in the anterior of the mesoderm at layers 4, 5, and 6 (4MA, 5MA, and 6MA) (Fig. 6D). In previously published seqFISH data of mouse embryos [77], we also observed that in E8.5 embryonic seqFISH sections, Irx1-positive cells, and Nr2f1-positive pSHF [64] cells converge into the CM from the anterior and posterior regions of SHF, respectively (Fig. 6E). Moreover, the activity of the Irx1 regulon was also enriched in area of C2, suggesting its function in controlling the specification of this aSHF subcluster (Additional file 1: Fig. S15B). Additionally, we found that the Irx1-positive subpopulation consistently exhibited high aSHF module scores (Additional file 1: Fig. S15 C-D), further highlighting its potential significance in driving key functional processes within the aSHF population.

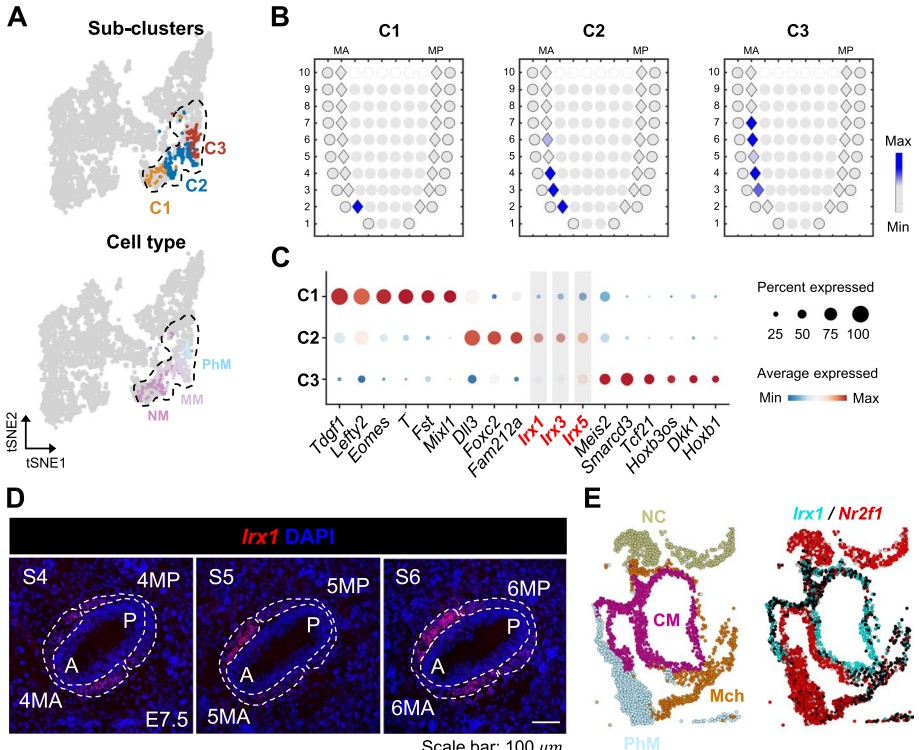

**Fig. 6** *Irx1* plays a crucial role in the development of the aSHF lineage. **A** E7.5 aSHF progenitors inferred by WOT analysis are re-clustered using the spatial feature genes. UMAP layout for the E7.5 aSHF progenitors is colored by subclusters (top) and cell type (bottom). The dashed line circles all aSHF progenitors at E7.5. **B** Corn plots showing the spatial pattern of inferred contributions of C1-3 clusters at E7.5. Color represents the estimated cell density of each cell type. **C** Dot plot showing the key markers of subclusters of E7.5 aSHF progenitors. Genes belonging to the Iroquois homeobox (*Irx*) family are marked in red. These genes were first identified as top-ranking genes using the FindMarker function in Seurat, based on LogFC and adj.pval calculations, and then manually selected considering their biological relevance in the context of our study. **D** RNAscope analysis showing that *Irx1* is highly specifically expressed in the anterior (A) region, particularly in layers 4/5/6. Scale bar: 100 μm. **E** Spatial positions, suggested by seqFISH data, of annotated single cells in the E8.5 heart section [77] (left) and normalized log expression counts of *Irx1* and *Nr2f1* (right). NC, neural crest

### *Irx1* is required for the development of the aSHF lineage and its derivatives

To trace these Irx1-positive cells in cardiac development, we generated an *Irx1*-reporter mouse model in which the *CreERT2-gapYFP or CreERT2-eGFP* cassette was targeted to the endogenous *Irx1* locus after the start codon (ATG) of *Irx1* (*Irx1-CreERT2-gapYFP or Irx1-CreERT2-eGFP*). *Irx1-CreERT2-gapYFP or Irx1-CreERT2-eGFP* mice were bred to *Rosa26-eYFP* or *Rosa26-tdTomato* (*Rosa26-tdT*) mice, respectively, and the pregnant female mice were subjected to tamoxifen injection for Cre-ERT2 activation at 6.25 ~ 6.5 dpc (Additional file 1: Fig. S16 A-B). The *Irx1-CreERT2; Rosa26-eYFP* or *Rosa26-tdT* embryos were then collected at E8.75 and E9.5 for the analyses of eYFP-positive or tdT-positive cells, which represent derivatives of Irx1-positive aSHF progenitors. We then quantified the contribution of eYFP/tdT-positive cells of *Irx1-CreERT2; Rosa26-eYFP* or *Irx1-CreERT2; Rosa26-tdT* embryos in the LV, RV, OFT, and aSHF where cells have not migrated into the developing heart. Contribution of the *Irx1*-lineage is the highest in aSHF at E8.75 and is higher in OFT than in RV and LV at E8.75-E9.5 (Fig. 7A, D, and E), suggesting Irx1-positive cells contribute

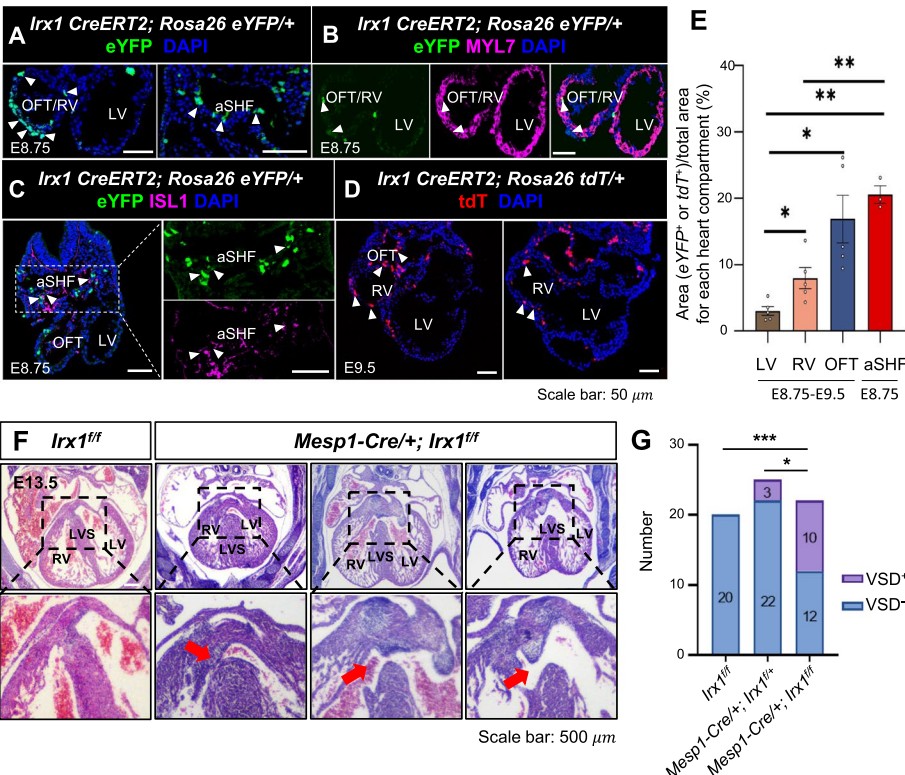

**Fig. 7** Deletion of *Irx1* in cardiac progenitors leads to ventricular septal defects in mice. **A** eYFP (green) and DAPI (blue) staining in embryonic heart sections at E8.75. Arrow heads indicate the contribution region of Irx1-positive cells at E8.75, including OFT and RV. **B** Immunofluorescence analysis showing the expression of MYL7 (ventricle marker) in the OFT/RV, atrioventricular canal (AVC), atrium (At), and endocardium (Endo). Arrowheads indicate areas where MYL7 and eYFP co-localize, highlighting the presence of Irx1-positive cells in these regions. **C** Immunofluorescence and lineage analysis revealing the co-localization of ISL1 (SHF marker) and eYFP in Irx1-positive cells. The arrowhead marks the specific cells expressing both *Isl1* and *Irx1*, highlighting their spatial distribution and potential role in cardiac development. **D** tdT (red) and DAPI (blue) staining in embryonic heart sections at E9.5. Arrow heads indicate the contribution region of Irx1-positive cells at E9.5, including OFT and RV. **E** Quantification of the contribution of the eYFP/tdT-positive cells in different heart compartments in E8.75 *Irx1-CreERT2; Rosa26-eYFP embryos and* E9.5 *Irx1-CreERT2;Rosa26-tdT* embryos. n represents the total number of embryos, with $n = 5$ in LV, RV and OFT at E8.75 and E9.5; and $n = 3$ in aSHF at E8.75. *P*-values were calculated using the Wilcoxon rank-sum test. Error bars are SEM. *$p < 0.05$, **$p < 0.01$. **F** Representative images of embryos at E13.5: one *Irx1$^{f/f}$* embryo on the left and three *Mesp1-Cre; Irx1$^{f/f}$* embryos on the right, with the latter exhibiting varying severities of ventricular septal defects. Red arrows indicate the locations of the ventricular septal defect in these embryos. Scale bar, 500μm. **G** Stacked bar chart showing the number of VSDs in *Irx1$^{f/f}$*, *Mesp1-Cre; Irx1$^{f/+}$* and *Mesp1-Cre; Irx1$^{f/f}$* embryos at E13.5. *n* is the number of embryos. The sample size (*n*) is as follows: n=20 for *Irx1$^{f/f}$*group, n=25 for *Mesp1-Cre; Irx1$^{f/+}$*group and n=22 for *Mesp1-Cre; Irx1$^{f/f}$*group. The number of embryos with or without VSD is directly labeled on the chart. *P*-values were calculated using the Chi-square test. *$p < 0.05$, ***$p < 0.001$.

more to the aSHF where they migrate into the developing heart through the OFT, than the primary heart field contributing to the LV. At E8.75, eYFP-positive cells colocalized with MYL7-positive CMs [78] in the OFT and RV (Fig. 7B), as well as in the remaining SHF colocalized with ISL1 [61] (Fig. 7C), further supporting the contribution of Irx1-positive cells to the aSHF lineage.

To examine the potential role of *Irx1* in the aSHF lineage, we generated the *Irx1$^{f/f}$* mice and bred with *Mesp1-Cre mice*, which allowed for mesoderm-specific *Irx1* deletion (*Irx1* CKO) (Additional file 1: Fig. S16 C-D). We then quantified the proportion of E13.5

*Irx1^{f/f}*, *Mesp1-Cre; Irx1^{f/+}*, and *Mesp1-Cre; Irx1^{f/f}* embryos exhibiting the VSD, characterized by the incomplete formation of the ventricular septum, with representative examples of *Mesp1-Cre; Irx1^{f/f}* embryos with VSDs of varying degrees of severity shown in Fig. 7F. Our analysis indicates that approximately 45.45% of *Mesp1-Cre; Irx1^{f/f}* embryos exhibit VSDs on E13.5, a proportion significantly higher than that observed in *Mesp1-Cre; Irx1^{f/+}* embryos, while no VSD is observed in *Irx1^{f/f}* embryos (Fig. 7G). These results demonstrate that deletion of *Irx1* in cardiac progenitors leads to high penetrance of the VSD.

These findings suggest that *Irx1* plays a pivotal role in aSHF development, significantly contributing to the formation of its derivatives, although its potential influence on broader mesodermal populations cannot be excluded. In summary, SEU-TCA serves as a powerful tool for integrating spatial and transcriptomic data, providing a foundational framework for studying development and disease.

## Discussion

Here, we introduce SEU-TCA, a novel computational framework specifically designed to integrate ST data with scRNA-seq data by utilizing TCA to obtain shared feature representations. Compared to other methods, SEU-TCA exhibits higher accuracy across multiple datasets, especially in spatial deconvolution and single-cell mapping tasks. By applying SEU-TCA, we have successfully constructed the dynamic differentiation process of early cardiac development at the single-cell level, identify three distinct cardiac progenitor lineages (JCF, aSHF, and pSHF), and resolve the spatial localization of these early cardiac progenitor cells. This demonstrates SEU-TCA's great performance in exploring the spatial distribution of progenitor cells during complex developmental processes.

After identifying three distinct cardiac progenitor lineages, we delve deeper into the aSHF lineage and identify possible distinct differentiation states of aSHF progenitors as early as E7.5. Our spatial deconvolution analysis using SEU-TCA reveals highly detailed spatial expression patterns of the *Irx* family genes, *Irx1*, *Irx3*, and *Irx5*, within the aSHF lineage. This is consistent with a previous report showing that these genes exhibit high homology and overlapping expression patterns throughout various stages of mouse embryonic heart development [74]. In comparison to these prior experiments, our analyses suggest the co-expression of *Irx1*, *Irx3*, and *Irx5* at an earlier time point and provide a more specific description of their co-expression within the aSHF lineage.

It has been well-documented that the double knockout of *Irx3* and *Irx5* results in abnormal orientation and arrangement of the OFT [79], yet the precise role of *Irx1* in cardiac development remained enigmatic. Prior studies on *Irx1* knockout mice have reported neonatal lethality primarily attributed to lung immaturity [80], and postnatal thinning of the compact layer of the ventricular wall [81], but these findings did not fully elucidate *Irx1*'s function in cardiac development. Building upon our SEU-TCA analysis, we have conducted genetic lineage tracing, which confirms that Irx1-positive cells contribute to aSHF's derivatives. Furthermore, through targeted deletion of *Irx1* in mesodermal cells using *Mesp1*-Cre, we have demonstrated that the absence of *Irx1* in the mesoderm leads to the VSD. These findings shed new light on the pivotal role of *Irx1* in cardiac development. Moreover, they underscore the ability of SEU-TCA to discern

intricate spatial expression patterns that may have previously eluded detection due to limitations in previous methodologies, thereby enhancing our understanding of the complex developmental landscape.

The superior performance of SEU-TCA stems primarily from its innovative approach to integrating spatially resolved transcriptomic data with scRNA-seq data. The key lies in the alignment of features within a shared latent space, achieved through the utilization of TCA. This alignment not only enables high-quality deconvolution of spatially resolved data but also facilitates accurate spatial mapping of scRNA-seq data. While SEU-TCA shares the goal of feature alignment with methods like SpaGE [19], it distinguishes itself through its fundamentally different approach. SpaGE relies on the PRECISE method, which involves performing independent Principal Component Analysis (PCA) on each dataset based on shared genes, thereby aligning spatial and scRNA-seq data into a shared latent space. However, this independent PCA approach inherently assumes linear relationships, potentially overlooking complex, non-linear structures prevalent in developmental datasets. In contrast, SEU-TCA minimizes the MMD between spatial and scRNA-seq data to obtain matched shared latent feature representations. As a non-parametric method, MMD not only leverages the kernel trick to map data into a high-dimensional feature space, making distribution differences easier to measure and minimize, but also allows for the selection of different kernels (e.g., Linear, Primal, RBF). This flexibility enables SEU-TCA to be tailored to the specific characteristics of the data, particularly when dealing with complex and diverse data distributions.

While SEU-TCA demonstrates robust performance, certain limitations should be acknowledged. Data preprocessing steps, such as normalization and feature selection, may introduce biases that affect spatial mapping accuracy. For instance, the selection of highly variable genes, while capturing key transcriptional variability, may overlook less-studied genes with potential biological relevance. Additionally, parameter choices, such as the removal of cell-spot pairs with low PCCs (PCC < 0.7), improve robustness but might exclude biologically meaningful associations with higher noise levels. Furthermore, the method's generalizability to datasets with different spatial resolutions, sequencing depths, or batch effects also requires further validation. Addressing these limitations in future studies will enhance the method's utility and interpretability.

Emerging single-cell spatial transcriptomics technologies, such as MERFISH [82] and Stereo-seq [83], offer high-resolution insights but they are often constrained by lower gene detection rates or prohibitive costs, limiting their feasibility for large-scale studies. In contrast, SEU-TCA leverages widely available multi-cell spot-based data to provide an effective computational framework for spatial transcriptomics analysis. This highlights its importance in bridging the gap between scalability and resolution in current spatial transcriptomics research.

## Conclusions

SEU-TCA presents a robust and versatile computational framework for the integrative analysis of single-cell and spatial transcriptomic data. SEU-TCA demonstrates superior performance across diverse tissues and developmental stages, underscoring its broad

applicability. Applying SEU-TCA to early cardiac development identified IRX1 as a critical spatially regulated transcription factor in aSHF specification, with genetic analyses validating its essential role in proper heart morphogenesis. These findings demonstrate SEU-TCA's capability to enhance spatial resolution in transcriptomic studies and reveal mechanistic insights into spatiotemporal gene regulation during embryogenesis.

## Methods

### Mice

All experiments involving animals were conducted in accordance with the NIH Guide for the Use and Care of Laboratory Animals. All animal protocols were approved by the Animal Care and Use Committee of Southeast University and the Institutional Animal Care and Use Committee (IACUC) of Tongji University. The mice were caged under SPF level conditions with 12 h light/dark cycles and given water and food and monitored daily for health.

*Irx1-creERT2-gapRFP/+* mice were purchased from GemPharmatech (Nanjing, China). *Irx1-creERT2-eGFP/+* mice were purchased from Cyagen Biosciences (Suzhou, China). *Rosa26-eYFP/Rosa26-eYFP* mice were originated from the Jackson Laboratory and were gifted from Pengfei Sui's Lab in Center for Excellence in Molecular Cell Science. *Rosa26-tdTomato/Rosa26-tdTomato* mice were originated from Fengchao Wang's Lab in Third Military Medical University (Chongqi, China). *Irx1-creERT2-eGFP/+* or *Irx1-creERT2-gapRFP/+* male mice were caged with *Rosa26-tdTomato/Rosa26-tdTomato* or *Rosa26-eYFP/Rosa26-eYFP* female mice, respectively, to generate *Irx1-creERT2-eGFP/+; Rosa26-tdTomato/+* (Irx1-creERT2; Rosa26-tdT) and *Irx1-creERT2-gapRFP/+; Rosa26-eYFP/+* (*Irx1-creERT2; Rosa26-eYFP*) mice. Pregnant females were identified with vaginal plugs in the following morning (E0.5) and were intraperitoneally injected with tamoxifen at a dose of 0.1 mg/g at 6.25 ~ 6.5 days post-coitum (d.p.c.).

*Irx1$^{f/+}$* mice were purchased from Cyagen Biosciences (Suzhou, China). *Mesp1*-Cre mice were originated from RIKEN Institute (RBRC01145) and were gifted from Zhongzhou Yang's Lab in Nanjing University. *Mesp1-Cre/+* and *Irx1$^{f/+}$* mice were mated to generate *Mesp1-Cre/+; Irx1$^{f/+}$* and *Mesp1-Cre/+; Irx1$^{f/f}$* embryos.

Mouse genomic DNA was obtained from mouse tail using alkaline lysis method. Embryonic genomic DNA was obtained from the tail and yolk sac of mouse embryos using Mouse Direct PCR Kit (Bimake, B40013; YEASEN, 10185ES70). PCR reaction was used to identify the genotype of the mice. Primer sequences used in this study are listed in Additional file 5: Table S4.

### RNAscope

RNAscope in situ hybridization of E7.5 mouse embryos was performed as previously reported [5]. Briefly, RNAscope analysis of *Irx1* was performed using RNAscope® Multiplex Fluorescent Reagent Kit v2 (Advanced Cell Diagnostics, 323,100) using probes supplied by Advanced Cell Diagnostics: mm-Irx1.

### Immunofluorescence

Pregnant mice were euthanized with $CO_2$ and 8.75 and 9.5 d.p.c. embryos were dissected using a pair of precise forceps. Mouse embryos were fixed with 4% PFA for 2 h at room temperature, washed with PBS, and saturated in a 30% sucrose solution before embedded in OCT. The embedded embryos were sectioned at 10 μm with a cryotome. To perform IF, wash the sliced samples with PBS then soak in 0.3% TritonX-100 for 40 min to penetrate the membrane. After washing the slices with PBS, block the sliced samples with ReadyProbes 2.5% Normal Goat Serum (Thermo) for 1 h and incubate overnight with diluted primary antibody at 4 ℃. Wash the sections with PBS and incubate the diluted secondary antibody at room temperature for 1–2 h. The embryo slices were washed again with PBS and were stained with DAPI at room temperature for 10 min, washed with PBS, sealed, and then imaged in a confocal microscope (LSM710, CarlZeiss).

### Antibodies

Antibody against MYL7 (Santa Cruz, sc-365255) (IF for embryos:1:100) was purchased from Santa Cruz. Antibody against ISL1 (Santa Cruz, sc-390793) (IF for embryos: 1:100) was purchased from Santa Cruz. Goat anti-mouse IgG(H +L) Highly Cross-Adsorbed Secondary Antibody, Alexa Fluor™ Plus 647(A32728) were purchased from Thermo Fisher Scientific.

### Hematoxylin and eosin (H&E) staining

Embryos at 13.5 d.p.c. were obtained using the same method mentioned above. Fresh embryos were fixed in 4% PFA overnight at 4 ℃, dehydrated using alcohol and vitrified in dimethylbenzene. Samples were then embedded in paraffin, sectioned at 6 μm and stained with H&E (GP1031; Servicebio Biotechnology, Wuhan, China). H&E staining was conducted according to the manufacturer's instruction. The stained slides were then imaged with a stereo microscope (Motic SMZ-171).

### Overview of SEU-TCA

SEU-TCA primarily consists of two components: TCA-based feature alignment and spatial mapping.

### TCA-based feature alignment

We employed TCA for feature alignment of scRNA-seq data from a spatially resolved transcriptomic data here. Specifically, TCA learns a good feature representation across domains via minimizing the discrepancy between two different distributions by mapping original data into the shared latent space.

In the SEU-TCA model, we denote $X_R$ as the reference spatial data with spatial position labels $Y_R$, and $X_Q$ as the single-cell query data missing spatial position labels $Y_Q$. The main goal of TCA is to find a fine transformation $\phi$ that can align $X_R$ with $X_Q$ in the new common latent feature space. From a statistical perspective, let $P(X_R)$ and $Q(X_Q)$ be the distributions of $X_R$ and $X_Q$, respectively, we expect that $P'\left(\{\phi(x_{R_i})\}\right) \approx Q'\left(\{\phi(x_{Q_i})\}\right)$.

Referring to domain adaption, we then utilized MMD between transformed reference spatial data $X_{R'} = \phi(x_R)$ and transformed scRNA-seq query data $X_{Q'} = \phi(x_Q)$ to measure the distance between these two distributions, the empirical estimate of the distance is written as:

$$Dist\left(X_{R'}, X_{Q'}\right) = \|\frac{1}{n_1}\sum_{i=1}^{n_1}\phi\left(x_{R_i}\right) - \frac{1}{n_2}\sum_{i=1}^{n_2}\phi\left(x_{Q_i}\right)\|_{\mathcal{H}}^{2}$$

Here $\mathcal{H}$ is defined as the universal Reproducing Kernel Hilbert Space (RKHS). $n_1$ and $n_2$ are the number of cells/spots in the corresponding dataset. Therefore, by mapping the distance between the means of the two datasets into the RKHS, the distance between the two datasets distributions can be accurately estimated.

Then, SEU-TCA seeks to minimize this distance governed by the nonlinear mapping $\phi$. To find a lower-dimensional representation, SEU-TCA uses a kernel function (Linear, Primal, RBF) to map the gene expression data into a low-dimensional space. Next, compute the Gram matrices $K_{R,R} K_{Q,Q}$ and $K_{R,Q}$ on reference, query, and cross-reference-query data, respectively, and combine them into a single kernel matrix $K$:

$$K = \begin{bmatrix} K_{R,R} & K_{R,Q} \\ K_{Q,R} & K_{Q,Q} \end{bmatrix}$$

Then construct the weighted matrix $L$, which satisfies $L_{ij} = 1/n_1^2$ if $x_i, x_j \in X_R$, else $L_{ij} = 1/n_2^2$ if $x_i, x_j \in X_Q$, otherwise $L_{ij} = -(1/n_1 n_2)$, and a centering matrix $H$ defined as:

$$H = I - \frac{1}{n}11^T$$

where $I$ is the identity matrix, 1 is a column vector of ones, and $n = n_1 + n_2$ is the total number of cells and spots. The optimization process minimizes the MMD between the transformed reference and query datasets in a shared latent space. Mathematically, the optimization problem is:

$$min_A tr\left(A^T KLKA\right) + \lambda tr\left(A^T A\right)$$

$$s.t. A^T KHKA = I$$

where $K$ is kernel matrix computed using the selected kernel function (Primal, Linear, or RBF). $L$ is weight matrix capturing domain discrepancy. The centering matrix $H$ is used to ensure that the kernel matrix $K$ is centered. $\lambda > 0$ is regularization parameter. $A$ is transformation matrix to project the data into a lower-dimensional shared latent space.

The optimization problem is solved through generalized eigenvalue decomposition of the matrices $KMK^T + \lambda I$ and $KHK^T$. The top $d$ eigenvectors derived from this decomposition form the transformation matrix $A$, which is subsequently used to project the reference and query data into the aligned feature space:

$$Z = \begin{bmatrix} Z_R \\ Z_Q \end{bmatrix} = KA.$$

### Kernel function selection

The SEU-TCA implementation supports three kernel types, which determine how the data is mapped into the latent space:

- Primal kernel: Directly uses the original data without mapping to a higher-dimensional space. Suitable for linearly separable data.
- Linear kernel: Computes pairwise dot products to capture linear relationships.
- RBF kernel: Maps data into a high-dimensional space to capture non-linear relationships. Its performance depends on the kernel bandwidth ($\gamma$).

The choice of kernel function significantly impacts the alignment quality. For example, the Primal kernel is computationally efficient but may not effectively align complex distributions. The RBF kernel is more flexible for non-linear distributions but requires tuning the $\gamma$ parameter.

### Spatial mapping

SEU-TCA utilizes the aligned TC representations to compute the PCC between the transformed spatial spots and the query cells. Specifically, for each query cell, the PCC is calculated with every spatial spot based on their respective TC profiles. This approach evaluates the linear relationship between the TC features of spatial spots and those of single-cell query data.

The spot-cell pair with the highest PCC value is determined to be the most likely spatial match for the query cell, indicating the closest similarity between their aligned TC profiles. To improve the reliability of the mapping and minimize noise, spot-cell pairs with PCC values below a predefined threshold (e.g., 0.7) are filtered out. This threshold acts as a quality control filter, ensuring that only mappings with sufficiently high statistical confidence are retained.

By adopting this approach, SEU-TCA effectively identifies high-confidence spatial mappings while minimizing the inclusion of spurious or ambiguous matches, thereby providing a reliable framework for integrating spatial transcriptomics and single-cell RNA-seq data.

### Benchmark comparison among different methods

We compared SEU-TCA with two mapping methods (Tangram [15] and SpaGE [19]) and four deconvolution methods (CARD [10], cell2location [9], STRIDE [11], and CIBERSORTx [23]). For all methods, we followed the tutorials available on their corresponding GitHub repositories. Moreover, the default parameter settings provided by these tutorials were adopted to ensure the consistency and standardization of the analysis process.

### Statistical analysis of performance metrics

To compare the performance of SEU-TCA with other methods, we conducted statistical analyses for the PCC metric and provided descriptive statistics for additional metrics (ACC, F1 score, Sensitivity, and Specificity).

For each method, the PCC was calculated to measure the consistency between mapped and true gene expression profiles. To demonstrate the uncertainty, we computed the mean PCC and its 95% confidence intervals (CIs) using standard methods based on the sample mean and variance. Pairwise comparisons between methods were conducted using *t*-tests. For each pair of methods, we calculated *p*-values and adjusted them for multiple testing using the False Discovery Rate (FDR).

All statistical analyses were conducted in R (version 4.1). Confidence intervals were calculated using standard error-based methods. Pairwise *t*-tests were conducted using the "t.test" function implemented in the stats package, and FDR-adjusted *p*-values were computed with the "p.adjust" function implemented in the stats package.

### Pseudo-bulk reconstruction ST for the human heart dataset

To generate pseudo-bulk spots from single-cell MERFISH spatial data, we utilized the grid-based approach to aggregate cells into spatially defined pseudo-spots. The following steps outline the procedure: (1) Grid assignment: Each single cell was assigned to a specific grid based on its spatial coordinates. The grid size was set to 100 units, and the row and column indices of each cell's corresponding grid were calculated. (2) Grid identification: A unique identifier was generated for each grid by combining the row and column indices. (3) Dominant cell type assignment: The cell type with the highest proportion within each grid was assigned as the ground truth label for the pseudo-spot. This process ensured that pseudo-bulk spots were accurately spatially aligned and correctly labeled with the dominant cell type, establishing a reliable ground truth for downstream analysis and benchmarking.

### Clustering analysis of spatial spots

Geo-seq data was processed using the Scanpy library (v1.8.2) in Python. At first, PCA was employed to reduce the dimensionality of the dataset while preserving the biological information. Then we select the top 20 principal components for downstream analysis, for example, construction of the neighborhood graph of spots, which was used for the clustering analysis using the Leiden algorithm. The resolution parameter of Leiden algorithm was set to 1 to balance the granularity of the clusters.

### Identification of spatial markers for three germ layers for the mouse gastrulation dataset

After spatial mapping by SEU-TCA, each cell in scRNA-seq data was assigned a spatial position label. We then identified DEGs for each spatial spot as markers using the Wilcoxon rank-sum test implemented by function "FindAllMarkers" in the R package Seurat. We selected top three markers for each spatial spot based on fold change, to visualize the consistency of single-cell and spatial data.

### Spatial regulon analysis

Here, we applied SCENIC pipeline to infer TFs and the gene regulatory network (regulon) in our single-cell transcriptome data with spatial position labels. The procedure mainly comprises three steps: (1) gene co-expression network construction: we utilized the GRNBoost algorithm to construct a gene co-expression network from the pre-processed gene expression matrix with default parameter settings. (2) Regulons, defined as TFs and their corresponding predicted target genes, were identified by RcisTarget. This step involves scanning for enriched TF binding motifs within the co-expressed gene sets identified by GRNBoost. (3) We quantified the activity of each regulon for each cell in spatial spots with AUCell. It assigns an enrichment score to each regulon based on the expression levels of its target genes, allowing for the identification of active regulatory programs in individual spatial locations. The regulon activity for each spatial location was defined as the average activity of the corresponding cells within that spatial location.

### Construction of the time-series tSNE maps for mesodermal lineage

The process of constructing a time-dependent tSNE map is divided into the following steps: (1) Input the data from two consecutive developmental stages, such as E6.75 and E7.0, using the former as reference and the latter as query. (2) Perform PCA on the reference and query data. (3) Use the "TSNE" function from the Python package openTSNE (v 1.0.1) to initialize and fit a tSNE model to the first 20 PCs of the reference data, with the parameters "initialization = pca", "exaggeration = 4", and "metric = cosine". (4) Calculate the correlation between the reference and query data across the variable genes from the reference. Identify the top $k$-nearest neighbors (kNN) for each query data point and assign each query data point to the median position of its nearest neighbors in the reference tSNE space. (5) Set the positions of the query data in the reference tSNE space as the initialization for the "TSNE" function, and finally generate a tSNE plot for the query data that aligns with the reference data. By analogy, the data at each subsequent time point utilizes the tSNE space of the previous time point as a reference to construct a time-dependent tSNE map.

### Cell type annotation for cardiac clusters

We selected the cardiac-associated cells at E8.5, including JCF progenitors (Mab21l2-positive Mch), SHF progenitors (Isl1-positive PhM), and mature CM (Nkx2-5-positive CM). We then applied the standard Seurat workflow to re-cluster these cardiac-associated cells and assigned cell type to each cluster according to its corresponding markers.

### Tracing the JCF, aSHF and pSHF lineages

Each lineage shown in Fig. 5 was inferred from a predefined starting cell set using the WOT analysis, as implemented in the Python package wot (v1.0.8) [58]. Specifically, E8.5 JCF cells were utilized for tracing the JCF lineage, E8.5 aSHF cells for the aSHF lineage, and E8.5 pSHF cells for the pSHF lineage. To ensure the overall quality of the tracing, we only kept cells with WOT score greater than 0.0001 for each lineage. To

further determine the lineage of each cell, we assigned it to the lineage corresponding to its highest WOT score. The WOT was utilized with the default parameters setting as in the Waddington-OT online tutorial (https://broadinstitute.github.io/wot/tutorial/).

### Sub-clustering of aSHF progenitors

Here, we performed sub-clustering analysis for aSHF progenitors at E7.5, which was predicted by WOT. The procedure is as follows. First, the function "FindAllMarkers" in R package Seurat was applied to identify DEGs on aSHF progenitors with spatial position labels at E7.5. Next, we performed PCA with the function "RunPCA" implemented in R package Seurat, where the parameter feature is set to the union of the top 20 DEGs for each spatial location. At last, we performed clustering analysis with the function "FindNeighbors" and "FindClusters" implemented in R package Seurat on the top 30 principal components generated by PCA, where the parameter resolution was set to 1. Here, we re-clustered the E7.5 aSHF progenitors, based on the spatial feature genes derived from the ST atlas, and achieved five clusters. However, two clusters were characterized by SM markers and Mch markers, respectively, so we excluded them from further analyses

### Supplementary Information

---

Additional file 1. Supplementary figures.

Additional file 2: Table S1. Highly region-specific genes for E7.5 mesodermal cells.

Additional file 3: Table S2. Average of RAS by each spot from spatial regulon map.

Additional file 4: Table S3. Top 30 marker genes for each of the three progenitor clusters (JCF, aSHF, and pSHF) at E8.5.

Additional file 5: Table S3. Primers used in the current study.

Additional file 6: Peer review history.

---

### Acknowledgements
The authors are grateful to the Lin & Luo lab members for helpful discussion of this study. We thank Prof. Pengfei Sui and Prof. Fengchao Wang for providing the *Rosa26-eYFP/Rosa26-eYFP* and *Rosa26-tdTomato/Rosa26-tdTomato* mice, respectively. We thank Ms. Qingyun Pan for technical assistance.

### Peer review information
Shila Ghazanfar and Wenjing She were the primary editors of this article and managed its editorial process and peer review in collaboration with the rest of the editorial team. The peer-review history is available in the online version of this article.

### Authors' contributions
C.L. and P.X. designed the research. J.H. and Y.Y. analyzed the data. R.J., H.C., Y.Z., X.J., and X.X. performed the *Irx1* lineage tracing and CKO phenotype analysis. X.Y. and N.J. performed the *Irx1* RNAscope experiment. Z.Y. designed the *Irx1* lineage tracing mice. J.H., Y.Y., P.X., Z.L., and C.L. wrote the manuscript. C.L., P.X., K.W., and Z.L. supervised the project. All authors read and approved the final manuscript.

### Funding
Studies in this manuscript were supported by funds provided by National Key R&D Program of China (2018YFA0800100 and 2018YFA0800101 to C.L.; 2018YFA0800103 to Z.L. and P.X.; 2018YFA0800104 to K.W.), the National Natural Science Foundation of China (32030017 to C.L.; 32100529 to P.X.; 32070823 to K.W.).

### Data availability
The previously published scRNA-seq data from mouse gastrulation that were re-analyzed here are available under accession codes E-MTAB-6967 from ArrayExpress [1, 84]. The previously published Geo-seq data that were re-analyzed here are available under accession codes GSE120963 [4, 5, 85]. Data from the human heart dataset is available for download from the Dryad Digital Repository [22, 86]. Data from the mouse olfactory bulb dataset is available for download from the Spatial Transcriptomics Research website [24, 87]. Data from the pancreatic ductal adenocarcinoma dataset is available from GEO under accession code GSE111672 [26, 88]. All other data supporting the findings of this study are available from the corresponding author on reasonable request. The SEU-TCA algorithm and source code used in this study are available

at GitHub [89] and archived at Zenodo [90]. The code is released under the MIT License. All immunofluorescence and hematoxylin and eosin (H&E) staining images used in this study have been deposited in Zenodo [91].

## Declarations

### Ethics approval and consent to participate
All animal experiments were approved by the Animal Care and Use Committee at Southeast University and the Institutional Animal Care and Use Committee (IACUC) of Tongji University, and performed in accordance with institutional guidelines.

### Consent for publication
Not applicable.

### Competing interests
The authors declare no competing interests.

### Author details
[1]Department of Cardiac Surgery, Key Laboratory of Developmental Genes and Human Disease, School of Life Science and Technology, Zhongda Hospital, Southeast University, Nanjing, China. [2]Co-Innovation Center of Neuroregeneration, Nantong University, Nantong, China. [3]Institute for Regenerative Medicine, State Key Laboratory of Cardiology and Medical Innovation Center, Shanghai Key Laboratory of Signaling and Disease Research, Frontier Science Center for Stem Cell Research, School of Life Sciences and Technology, Shanghai East Hospital, Tongji University, Shanghai, China. [4]Guangzhou Laboratory, Guangzhou, China. [5]State Key Laboratory of Pharmaceutical Biotechnology, Department of Cardiology, Model Animal Research Center, Nanjing Drum Tower Hospital, The Affiliated Hospital of Nanjing University Medical School and MOE Key Laboratory of Model Animal for Disease Study, Nanjing University, Nanjing, China. [6]State Key Laboratory of Digital Medical Engineering, School of Biological Science & Medical Engineering, Southeast University, Nanjing, China.

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

## 