## [Additional file 6: Peer review history. · Genome Biology]

Review history

First round of review

Reviewer 1

This manuscript describes SEU-TCA, a new method for integrating single-cell RNA sequencing (scRNA-seq) and spatial transcriptomics (ST) data. The authors apply this method to investigate the spatial organization and lineage relationships of early cardiac progenitors during mouse embryogenesis. They use publicly available GEO-seq and scRNA-seq datasets from E7.5 and E8.5 mouse embryos to map different cell types, identify spatial regulons, and trace the developmental trajectories of cardiac lineages. They further validate their findings through experimental studies, including lineage tracing and gene knockout experiments using a mouse model with conditional deletion of the *Irx1* gene.

I believe it offers methodological value by demonstrating how existing knowledge can be integrated with spatial techniques to investigate the regulation and spatial distribution of progenitor pools during embryogenesis. While the study does not represent a major paradigm shift in how cardiac fields are deployed, it provides a robust resource for researchers and enhances prior lineage tracing and scRNA-seq knowledge. The methodological innovation of SEU-TCA also shows promise for broader application in developmental biology, making it a useful contribution to embryogenesis research.

Main strengths of the method:

- It enables greater accuracy and deconvolution of spatial data and prediction of single-cell locations than previous methods.
- The authors successfully identify the three distinct cardiac lineages as classified to date-juxta-cardiac field (JCF), anterior second heart field (aSHF), and posterior second heart field (pSHF)- and corroborate their distinct spatial localization at E7.5. This can be considered a solid proof of concept.
- The experimental validation of *Irx1* function using lineage tracing and conditional knockout mouse models strengthens the biological relevance of the computational findings. *Irx1* is expressed in aSHF progenitors and its deletion leads to ventricular septal defects, showing that SEU-TCA can lead to relevant discoveries.

Main points to address:

1. The manuscript could benefit from a more thorough explanation of the mathematical basis of the method. A more detailed description of the TCA algorithm, including its optimization process, the selection of kernel functions (linear, primal, RBF), and how SEU-TCA minimizes the MMD between the data distributions would improve the manuscript. For example, clarify how the selection of specific kernel functions influences the alignment process and the resulting spatial mapping. Provide details on kernel parameters, their optimization, and robustness across datasets. The github page only provides a succinct jupyter notebook "SEU-TCA.ipynb". Authors should consider adding a more documented/commented version or uploading extensive documentation. That would allow other researchers to adapt the method to other types of spatial datasets.

Thank you for your insightful comments. We have revised the manuscript to provide a more comprehensive explanation of the mathematical foundation of SEU-TCA, including

the TCA algorithm, its optimization process, kernel selection, and parameter tuning. Below are the detailed responses to your concerns:

1. Optimization process in TCA algorithm

“The SEU-TCA method employs Transfer Component Analysis (TCA) to align distributions of scRNA-seq data and spatial transcriptomic data. The optimization process minimizes the Maximum Mean Discrepancy (MMD) between the transformed reference and query datasets in a shared latent space. Mathematically, the optimization problem is:

$$\begin{aligned} \min_A \quad & \text{tr}(A^T K L K A) + \lambda \text{tr}(A^T A) \\ \text{s.t.} \quad & A^T K H K A = I \end{aligned}$$

Where K is kernel matrix computed using the selected kernel function (primal, linear, or RBF). L is weight matrix capturing domain discrepancy. The centering matrix H is used to ensure that the kernel matrix K is centered. $\lambda > 0$ is a regularization parameter. A is transformation matrix to project the data into a lower-dimensional shared latent space. The optimization problem is solved through generalized eigenvalue decomposition of the matrices $K M K^T + \lambda I$ and $K H K^T$. The top d eigenvectors derived from this decomposition form the transformation matrix A , which is subsequently used to project the reference and query data into the aligned feature space:

$$Z = \begin{bmatrix} Z_R \\ Z_Q \end{bmatrix} = K A$$

”

We updated the manuscript to include these steps and their practical implementation, which closely follows the provided code.

2. Kernel function selection and its influence

“The SEU-TCA implementation supports three kernel types, which determine how the data is mapped into the latent space:

- *Primal kernel: Directly uses the original data without mapping to a higher-dimensional space. Suitable for linearly separable data.*
- *Linear kernel: Computes pairwise dot products to capture linear relationships.*
- *RBF kernel: Maps data into a high-dimensional space to capture non-linear relationships. Its performance depends on the kernel bandwidth (γ).*

The choice of kernel function significantly impacts the alignment quality. For example, the Primal kernel is computationally efficient but may not effectively align complex distributions. The RBF kernel is more flexible for non-linear distributions but requires tuning the γ parameter.”

We clarified how the kernel function is chosen based on data characteristics and its influence on the spatial mapping results. We also conducted extensive experiments on the human heart dataset using alignment dimensions ranging from 10 to 100 under the three kernel types (Primal, Linear, RBF). The evaluation metrics used included ACC, F1 score, Sensitivity, and Specificity. Our results demonstrated that dimensions above 30 had minimal impact on performance, and the Primal kernel consistently outperformed the Linear and RBF kernels in this dataset. These findings are illustrated in **Fig. S2**.

3. Kernel parameter optimization and robustness

To address the reviewer’s comment regarding kernel parameter optimization and robustness, we have conducted a comprehensive evaluation of the bandwidth parameter (γ) for the RBF kernel. Specifically, we used grid search to examine γ values of 0.1, 1, and 10 on the human heart dataset to assess the robustness of the SEU-TCA method. Our findings demonstrated that the performance differences among these γ values were minimal, indicating that the method is robust to variations in this parameter within a reasonable range. These results are presented in detail in **Fig. S2**.

4. Update GitHub page

We agree that the importance of detailed documentation in helping researchers apply SEU-TCA to various spatial datasets. To address this, we have updated the GitHub repository with a more thoroughly commented version of the SEU-TCA notebook (<https://github.com/LinluoLab/SEU-TCA/blob/master/README.md>), explaining the theoretical background, parameter settings, and practical implementation details. Additionally, we have revised the user manual, including a step-by-step guide, kernel parameter optimization strategies (e.g., γ for the RBF kernel), and examples of applying SEU-TCA to different datasets. A quick start guide has also been provided to help users get started easily. These updated resources are now available at <https://github.com/LinluoLab/SEU-TCA>. We hope these improvements address your concerns and support the research community in applying SEU-TCA effectively.

2. The manuscript states SEU-TCA performs better than Tangram and SpaGE. However, benchmarking results (Additional File 1: Fig. S2) do not include uncertainty estimates (e.g., confidence intervals). Provide statistical validation of comparative performance. Also, when multiple comparisons are performed, it is important to adjust the p-values to control for the increased likelihood of false positives (Type I errors). Using a raw p-value from a t-test without adjustment can lead to incorrect conclusions (For example, if you perform 20 independent tests with a significance threshold of 0.05, there is a ~64% chance of getting at least one false positive). You can use one of the multiple methods for p.value adjustment (Bonferroni Correction, FDR, Holm's Step-Down or permutation-based methods) or opt for ANNOVA.

Thank you for your thoughtful feedback and for highlighting important aspects of the statistical validation. We have carefully addressed your concerns as follows:

For the Pearson Correlation Coefficients (PCC) metric presented in **Additional File 1: Fig. S7A**, we recalculated the results across the original dataset and three additional datasets. To account for uncertainty, we computed 95% confidence intervals for the PCC values of each method. Pairwise statistical comparisons between methods were conducted using t-tests, and we applied the False Discovery Rate (FDR) method for multiple testing correction to mitigate the risk of Type I errors.

We have incorporated detailed descriptions of these analyses in the revised manuscript, including updates to the **Methods section**, **Results section**, and **Figure legends**.

3. Marker selection is crucial for identifying spatially restricted populations, especially when using non-spatial scRNA-seq data. The fuzzy boundaries between aSHF and pSHF further emphasize this. The referencing and explanation on the text could be extended on this sense. For example “*Foxc2* and *Hoxb1*, which are early markers for aSHF and pSHF lineages, respectively [60]” reference 60 does not show primary evidence of the fact that *Foxc2* is restricted to the aSHF, but utilizes this as a marker, instead, authors could reference the first primary evidence that links *Foxc2* to the aSHF and OFT development (<https://doi.org/10.1016/j.ydbio.2006.06.012>). Make sure this point is revised for all the marker literature referencing, cite the original paper that described it and not sc-atlases that use it. Similarly, Figure 4C does include *Fgf8* as a marker for aSHF, but it is not discussed or referenced in the text. *Tcf21* is mentioned in the context of chick development but not specifically for mouse (or at least not clearly specified). Given the importance in distinguishing aSHF in this study, using a broader set of markers would strengthen the ground of the following *in vivo* validation. A suggestion (but not requirement) would be using Seurat's AddModuleScore function. This function allows you to assess the expression of a set of genes (a module) within each cell. That way you could validate that your *Irx1+* subpopulation is indeed within the aSHF. Confirming whether they exhibit high scores for the aSHF module would provide further evidence for their identity.

Thank you for pointing out the importance of using original references for marker selection and the need for clarification regarding the markers discussed in the manuscript. We have addressed your concerns as follows:

1. Revising references for *Foxc2* and *Hoxb1*

We acknowledge that Reference 60 does not provide the primary evidence linking *Foxc2* to the aSHF or *Hoxb1* to the pSHF. We have replaced this reference with the original study that first described *Foxc2*'s role in aSHF and OFT development (DOI: [10.1016/j.ydbio.2006.06.012](https://doi.org/10.1016/j.ydbio.2006.06.012)). Similarly, for *Hoxb1*, we have reviewed and cited the

primary evidence that supports its association with the pSHF (DOI: 10.1016/j.ydbio.2011.02.029 and 10.7554/eLife.55124).

We have conducted a thorough review of all marker references in the manuscript to ensure that they cite the original studies that first described these markers, rather than relying on secondary references or scRNA-seq atlases.

2. Comprehensive annotation of aSHF markers

We have revised the manuscript to provide a more detailed discussion of markers for the aSHF, incorporating *Fgf8* as a key marker and referencing primary evidence supporting its role in SHF progenitor specification and outflow tract development. Additionally, we have included *Tcf21*, along with *Isl1* and *Tbx1*, to present a broader and more robust set of aSHF markers. These updates underscore the accuracy of our aSHF annotation and provide a stronger basis for subsequent in vivo validation. Furthermore, we have expanded the discussion of the JCF and pSHF markers, ensuring consistent and comprehensive coverage of all major cardiac progenitor clusters. The updated text now reads as follows:

*“The JCF is defined by the expression of *Mab21l2* [60] and *Hand1* [62], while the pSHF exhibits enriched expression of markers including *Osr1* [63] and *Nr2f2* [64], reflecting its posterior spatial identity and lineage commitment. The aSHF, on the other hand, is characterized by genes such as *Isl1* [61], *Tbx1* [65], *Fgf8* [66], and *Tcf21* [38], which collectively underscore its pivotal roles in second heart field progenitor specification and outflow tract (OFT) development (Fig. 5C; Additional file 4: Table S3).”*

To further support our findings, we also have identified and updated the top 30 marker genes for each of the three progenitor clusters (JCF, aSHF, and pSHF) in the **Additional file 4: Table S3**.

3. Validation of Irx1-positive subpopulation identity using aSHF module scores

We agree that using Seurat’s AddModuleScore function to validate the aSHF identity of the Irx1-positive subpopulation could provide additional support for our findings. To address this, we calculated module scores for a curated set of aSHF marker genes (e.g., *Isl1*, *Tbx1*, *Fgf8*, *Tcf21*) across all cells in our dataset. This analysis revealed that the Irx1-positive subpopulation consistently exhibits high module scores, further confirming its classification within the aSHF.

We have added these results to the revised manuscript (**Fig. S15D**, shown below) and included a detailed description of the analysis in the **Methods** and **Results** sections. This additional evidence strengthens our conclusion that the Irx1-positive subpopulation belongs to the aSHF.

4. The functional relevance of the *Irx1* regulon is validated through direct perturbation of *Irx1* in vivo using *Mesp1-Cre*; however, critical points are missing:

a. Sample size, severity and penetrance: The manuscript does not report the number (n) of embryos analyzed, the penetrance, or the proportion of embryos displaying ventricular septum defects as shown in figure 5E. Consider including a supplementary figure with representative examples of mutant embryos exhibiting varying severities of defects to determine whether the phenotypes range along a spectrum (e.g., mild, moderate, or severe) or are binary (affected/unaffected) and a quantification of these phenotypes. This is critical to address the role of *Irx1*.

Thank you for your valuable comment on our manuscript. We displayed three representative examples of *Mesp1-Cre; Irx1^{ff}* embryos with ventricular septal defects (VSDs) of varying degrees of severity (as shown in Fig. 7F) and quantified the proportion of E13.5 *Irx1^{ff}*, *Mesp1-Cre; Irx1^{+/+}*, and *Mesp1-Cre; Irx1^{ff}* embryos exhibiting VSD (as shown in Fig. 7G). Our analysis indicates that approximately 45.45% of *Mesp1-Cre; Irx1^{ff}* embryos exhibit ventricular septal defects on E13.5, a proportion significantly higher than that observed in *Mesp1-Cre; Irx1^{+/+}* embryos, while no VSD is observed in *Irx1^{ff}* embryos (as shown in Fig. 7G). These results demonstrate that deletion of *Irx1* in cardiac progenitors leads to high penetrance of VSD.

b. The lineage tracing experiment using *Irx1*-CreERT2 (Figure 5F-I) would benefit from quantification to provide a more comprehensive understanding of *Irx1* expression and specificity. While representative images are presented, quantifying the proportion of tdTomato-positive cells in each heart chamber (e.g., left ventricle, right ventricle, and outflow tract) is required. Determining the amount of tdTomato contribution to the left ventricle (LV), which primarily derives from the first heart field (FHF), would help evaluate how specific *Irx1* is to second heart field (SHF) lineages.

Thank you for the important suggestion. We quantified the contribution of tdTomato-positive cells of *Irx1*-CreERT2-*eGFP*; *Rosa26*-*tdT* embryos in the LV, RV, OFT, and aSHF where cells have not migrated into the developing heart. As shown below in Fig. 7E, contribution of the *Irx1*-lineage is the highest in aSHF at E8.75, and is higher in OFT than in RV and LV at E8.75-E9.5, suggesting *Irx1*-positive cells contribute more to the aSHF where they migrate into the developing heart through the OFT, than the primary heart field contributing to the LV. The relative low presence of *Irx1*-lineage cells to the RV may suggest that these cells belong to the later migrating aSHF cells than the earlier ones contributing to the RV, but it could also be a result of the timing of tamoxifen treatment.

c. *Mesp1*-Cre recombines across the anterior mesoderm, including both the first heart field (FHF) and second heart field (SHF), as well as non-cardiac mesodermal populations. The ventricular septum is known to derive, at least partially, from both FHF and SHF (i. e. a population at the border of the FHF and SHF *Tbx5*+/*Mef2c*AHF+ progenitor lineage; (<https://www.biorxiv.org/content/10.1101/2024.02.05.578995v2>). This raises the question of whether the observed phenotypes result specifically from disruption in aSHF progenitors or other mesodermal lineages. *In situ* hybridization at E8.75 of *Irx1* on both *Mesp1*Cre/+; *Irx1* fl/+ and *Mesp1*Cre/+; *Irx1* fl/fl would help clarify this.

Thank you for this important suggestion to our manuscript. Indeed, the *Mesp1*-lineage covers most the cells in the heart, and qRT-PCR showed that *Irx1* is almost completely deleted in the E9.5 *Mesp1*-Cre; *Irx1*^{fl/fl} embryos (Fig. S16D, shown below). As we have shown in the quantification of contribution of *Irx1* lineage cells in the response the last question, they contribute more to aSHF and OFT than to RV and LV, suggesting the VSD observed in *Mesp1*-Cre; *Irx1*^{fl/fl} embryos is likely to be attributed by defects in the aSHF. However, as *Irx1* is expressed later in the septum of the developing heart (DOI: 10.1006/dbio.2000.9801; 10.1016/s0925-4773(01)00451-8), the VSD may also be the result of loss of *Irx1* in the developing septum directly. More spatiotemporally precise Cre lines may be needed to distinguish the specific functions of *Irx1* in different stage and compartment of the developing heart.

5. Discussion of limitations: The authors should expand the discussion section to include a more critical assessment of the limitations of the SEU-TCA method. Addressing potential biases introduced by data preprocessing, feature selection, or the choice of parameters would improve the study's transparency and help readers interpret the results more cautiously. For example, the authors could discuss how the removal of cell-spot pairs with low Pearson correlation coefficients ($PCC < 0.7$) during spatial mapping might affect the results.

We appreciate the reviewer's insightful suggestion to provide a more critical assessment of the limitations of the SEU-TCA method. In response, we have expanded the discussion section to address potential biases introduced by data preprocessing, feature selection, and parameter choices. Specifically, we have included a discussion of the impact of filtering out cell-spot pairs with low Pearson correlation coefficients ($PCC < 0.7$) on the results, as suggested. This addition aims to improve the transparency of our study and helps readers interpret the findings with appropriate caution.

The expanded discussion section now reads as follows:

“While SEU-TCA demonstrates robust performance, certain limitations should be acknowledged. Data preprocessing steps, such as normalization and feature selection, may introduce biases that affect spatial mapping accuracy. For instance, the selection of highly variable genes, while capturing key transcriptional variability, may overlook less-studied genes with potential biological relevance. Additionally, parameter choices, such as the removal of cell-spot pairs with low PCCs ($PCC < 0.7$), improve robustness but might exclude biologically meaningful associations with higher noise levels. Furthermore, the method's generalizability to datasets with different spatial resolutions, sequencing depths, or batch effects also requires further validation. Addressing these limitations in future studies will enhance the method's utility and interpretability.”

Minor Points

1. “We applied SEU-TCA to analyze GEO-seq and scRNA-seq data for E7.5 (Late Streak stage)”. E7.5 would correspond to 0B-EB stage according to the currently most accepted staging system for gastrulating mouse embryos (10.1242/dev.118.4.1255). Late streak is more in between E7.0 and E7.25.

Thank you for pointing out the inconsistency in staging terminology. We acknowledge that the term “Late Streak stage” (E7.0–E7.25) does not strictly correspond to E7.5, as

E7.5 would align with the early “0B-EB stage”. To ensure accuracy, we have replaced “Late Streak stage” with “0B-EB stage” in the manuscript.

2. “during which the overall morphology and organizational structure start to manifest”. Consider rephrasing, it is not clear what you are refereeing to. You can take inspiration on how to define this critical embryo stage from elegantly written reviews such as [10.1016/j.mod.2020.103617](https://doi.org/10.1016/j.mod.2020.103617)

Thank you for your comment, we have revised the description to clarify the developmental context of the E7.5 (0B-EB stage) mouse embryos. The updated text now reads:

“To overcome the inadequacy of gastrula ST data resolution and the lack of spatial information in the annotated gastrula scRNA-seq atlas, we next applied SEU-TCA to analyze GEO-seq and scRNA-seq data for E7.5 (0B-EB stage) mouse embryos, a critical developmental period characterized by gastrulation, during which the embryonic germ layers are formed, and the body axes are established [27].”

3. The manuscript would benefit from proposing a plausible mechanistic insight. A potential way to address this would be:
 - a. Identify downstream effectors modulated by *Irx1* (e.g., genes within the *Irx1* regulon inferred by SCENIC). Are these effectors expressed specifically in the aSHF in your dataset? You may also validate their expression using RNA-FISH or immunostaining in *Mesp1-Cre Irx1* mutants. Alternatively, you could perform bulk RNA-seq on dissected hearts from mutant and control embryos to identify transcriptional changes. This would clarify whether *Irx1* impacts aSHF-specific genes or has broader effects on mesodermal development. Providing this information will strengthen the conclusion that *Irx1* functions specifically within the aSHF lineage.
 - b. A less time-consuming alternative would be to tone down the conclusions regarding the specificity of *Irx1* to the anterior second heart field (aSHF) and instead clarify that the effects of *Irx1* might not be specific to aSHF progenitors. It is important to acknowledge that the observed defects could result from disruption in both lineages or other mesodermal populations. Without quantification of lineage tracing data that determines the contribution of *Irx1* to different heart regions (e.g., left ventricle, right ventricle, aSHF), the manuscript should present these findings with more caution. Specifically, revising the conclusions to reflect the possibility that *Irx1* may influence broader mesodermal development rather than being strictly confined to aSHF. For example, in the text “This suggests that *Irx1* is functionally important for aSHF development. Thus, *Irx1* is specifically expressed in aSHF and is essential for the development of the aSHF lineage and the formation of ventricular septum” should be rephrased if no quantification or further experiments are provided.

We thank the reviewer for their insightful comments regarding the mechanistic insights into *Irx1* function. In our analysis, we identified *Foxc1* and *Foxc2* as downstream regulatory genes of *Irx1*, both of which are specifically expressed in the aSHF in our dataset. These findings align with prior studies that reported the critical roles of *Foxc1* and *Foxc2* in aSHF development (DOI: [10.1016/j.ydbio.2006.06.012](https://doi.org/10.1016/j.ydbio.2006.06.012)), but we have not yet tested the expression of these genes in *Irx1* mutant embryos to determine whether their regulation is directly impacted.

Additionally, we performed lineage tracing experiments and quantified the contribution of *Irx1*-positive progenitors to different cardiac regions. Our data show that *Irx1*-positive progenitors contribute significantly more to the aSHF and its derivatives, including the OFT and RV, compared to the FHF and its derivative, the LV. These findings support the preferential role of *Irx1* in the aSHF lineage.

However, we recognize that the observed phenotypes and molecular changes might not be exclusively restricted to the aSHF. It remains possible that *Irx1* could also influence broader mesodermal development. Therefore, we have revised the conclusions to present a more cautious interpretation. Specifically, we now clarify that while *Irx1* plays a significant role in aSHF development, its effects may extend beyond this specific lineage to other mesodermal populations. The updated text now reads as follows:

*“These findings suggest that *Irx1* plays a pivotal role in aSHF development, significantly contributing to the formation of its derivatives, although its potential influence on broader mesodermal populations cannot be excluded.”*

4. Embryo stage should always be specified in cornplots, either in the figure diagram or on the legend. Check whether it is like that on all figures. For example, legend “Figure 2F. Corn plots showing the spatial pattern of expression of 7MA markers and UMAP showing the single-cell resolution pattern of expression of 7MA markers.”, lacks this info.

Thank you for highlighting this essential requirement for our figures. We acknowledge that the embryo stage information was missing in some legends as you noted. Since all our cornplots are from the embryo at the E7.5 stage, we have promptly updated all relevant figure legends to explicitly state to ensure clarity and compliance with the requirement.

5. Clarity and precision in language: The manuscript would benefit from revisions to improve clarity and precision in language. Certain sentences are overly complex or ambiguous. Specific examples include:
 - "lacks the crucial spatial context, which is necessary for a comprehensive understanding of cellular dynamics within the gastrula" could be revised to "lacks the crucial spatial

context to fully understand cellular dynamics within the gastrula" for improved conciseness and clarity.

Thank you for your excellent suggestion. Your observation is entirely correct that the revised sentence is more concise and clear compared to the original one. We fully appreciate your efforts in helping us improve the clarity of our expression, and we have revised it in our manuscript.

- The informal phrase "To our knowledge, this is the first..." should be rephrased to maintain a formal tone consistent with a scientific publication. Consider replacing it with "This represents..." or a similar phrase that conveys the same meaning while maintaining formality.

Thank you for your valuable comments. We fully agree with you that our previous description was rather informal and might not be in line with the formal tone expected in a scientific publication. We have revised the sentence to read:

"To date, this study represents the first systematic spatio-temporal tracing of early cardiac developmental trajectories."

- "The regulon activity for each spatial location was defined as the average activity of the corresponding cell sets within a spatial location" can be made more concise by revising it to "The regulon activity for each spatial location was defined as the average activity of the corresponding cells within that location."

Thank you very much for your valuable suggestion. You are absolutely right that the revised sentence is much more concise while still maintaining the key meaning. We fully agree with this improvement and have revised it in our manuscript. Your careful attention to detail is truly appreciated and has helped enhance the clarity and readability of our work.

- "numerous previously unrecognized spot-specific regulons" is hard to read, many adjectives preceding a noun. Consider rephrasing.

Thank you for pointing out the readability problem. We agree that the current construction is difficult. We have restructured the sentence to read as follows:

"The spatial regulon map also revealed a large number of spot-specific regulons and TFs that had previously been unrecognized, which might be potential candidates for further functional validation."

We believe this revised version improves the flow of the text and enhances comprehension.

- "Additionally, the spatially mapped single-cells were used to generate pseudo GEO-seq profiles, which well recovered the original expression patterns" should be changed to "Additionally, the spatially mapped single-cells were used to generate pseudo GEO-seq profiles, which accurately recovered the original expression patterns."

I sincerely thank the reviewer for this valuable suggestion. The change from "well" to "accurately" indeed makes the description more precise and appropriate. We have promptly implemented this modification in the manuscript to ensure the highest quality of our expression.

- "These data suggest that cardiac lineage segregation initiates much earlier than the activation of marker genes, which poses a challenge for clarifying lineage origins using marker-based lineage tracing experiments." The conclusion that cardiac lineage segregation begins earlier than the activation of marker genes is intriguing. However, it

is important to note that cluster propagation to progenitor spaces using methods like optimal transport or maximum parsimony is not enough to propose distinct lineage identities. These computational approaches identify clusters or trajectories based on spatial or transcriptomic manifolds, which may reflect states of developmental plasticity rather than definitive lineage segregation. Functional validation, such as lineage tracing or fate-mapping experiments, would be necessary to substantiate the claim of early lineage segregation.

Thank you for your insightful comment. We agree that computational approaches alone are insufficient to definitively establish lineage identities and must be interpreted with caution. To address this, we propose that the differences in progenitor cell states we observe may reflect early lineage priming before the activation of traditional marker genes, rather than definitive lineage segregation. This idea aligns with the notion that progenitor cells in distinct spatial domains may exhibit subtle transcriptomic differences indicative of lineage bias, even in the absence of well-established markers. The revised text in the discussion now reads:

“Our findings suggest that progenitor cells from distinct spatial domains may begin to exhibit subtle transcriptomic differences, indicative of early lineage bias, even before the activation of canonical marker genes.”

- “Dysregulation or abnormalities in the development of the aSHF lineage have been associated with various congenital heart diseases (CHDs), including defects in the aorta, pulmonary artery, and ventricular septum [61, 62]. However, the underlying mechanisms remain unclear.” There is extensive literature proposing and validating mechanistic explanations for this aSHF to CHD links. Convergent extension, lack of migration, cell death, etc (<https://www.nature.com/articles/s41586-019-1414-x>, <https://www.ahajournals.org/doi/10.1161/CIRCRESAHA.115.305020>). “the underlying mechanisms remain unclear” is an over-simplification. Consider rephrasing.

We sincerely appreciate your perceptive observation and the valuable references you provided. We agree that the phrase “the underlying mechanisms remain unclear” might have been an over-simplification. To address this, we have rephrased the relevant sentence to better reflect the current state of knowledge and the complexity of the field. The revised sentence now reads:

“Dysregulation or abnormalities in the development of the aSHF lineage have been associated with various congenital heart diseases (CHDs), including defects in the aorta, pulmonary artery, and ventricular septum [72, 73]. However, the subclusters and key regulatory factors of aSHF require further in-depth exploration.”

We hope this revised wording provides a more accurate description of the research context.

- In Fig5C legend. “Dot plot showing the key markers of subclusters of E7.5 aSHF progenitors. Genes belonging to the Iroquois homeobox (Irx) family are marked in red”. Are these top-ranking genes on LogFC, adj. pval or manually selected genes? This should be clarified in the legend.

We sincerely appreciate your careful review and the valuable comment on the Fig5C legend. The genes presented in the dot plot of Fig5C were selected via a multi-step process. First, we employed Seurat’s FindMarker function to compute LogFC and adj.pval for ranking genes. Afterwards, we manually picked from this list of top-ranked genes, relying on our in-depth biological knowledge and the specific focus of our study

regarding the relevant biological processes. To address your concern, we've updated the legend of Fig5C as follows:

"Dot plot showing the key markers of subclusters of E7.5 aSHF progenitors. Genes belonging to the Iroquois homeobox (Irx) family are marked in red. These genes were first identified as top-ranking genes using the FindMarker function in Seurat, based on LogFC and adj.pval calculations, and then manually selected considering their biological relevance in the context of our study."

The revised figure along with its updated legend has been incorporated into the manuscript. Thank you for helping enhance the clarity of our figures and related descriptions.

- "The regulon activity for each spatial location was defined as the average activity of the corresponding cell sets within a spatial location." could be changed to "The regulon activity for each spatial location was defined as the average activity of the corresponding cells within that spatial location."

Thank you very much for your meticulous attention to the language and the valuable suggestion for improving this sentence. We have already updated the sentence in our manuscript exactly as you recommended. The modified sentence can now be found in its original position within the relevant section.

Reviewer 2

In this manuscript, J. He and colleagues introduce SEU-TCA, a novel computational framework de-signed to enhance the deconvolution of spatial transcriptomic datasets. By employing Transfer Component Analysis, the authors claim that SEU-TCA improves the alignment between single-cell RNA sequencing (scRNA-seq) data and spatial transcriptomics data, thereby enabling more accurate spatial mapping. They apply this algorithm to a GEO-seq dataset of a mouse gastrula, in the developmental context of cardiac progenitors and successfully identify precise progenitor lineages.

Given that my expertise is on the computational side, I will only focus on this side of the manuscript. My major concern is that, although SEU-TCA shows promise to be useful for the research community, given that the manuscript is presented as a methods development, a significant limitation is its lack of extensive benchmarking against other comparable computational approaches. Additionally, the method is only tested using a single dataset, making difficult to know if it would be generally applicable. I mention this and other minor concerns in a point-by-point manner below.

Major Points

- As mentioned above, the authors should include a more comprehensive evaluation, incorporating more computational approaches and a wider range of computational metrics. I also believe that given the authors make this a major point of the manuscript, this benchmark should be shown as a main figure. A good example of benchmarking using different metrics can be seen in the SPOT-light paper (Elosua et al 2021, NAR).

Thank you for your valuable feedback. In response to your suggestion to include a more comprehensive evaluation incorporating additional computational approaches and metrics, we have made significant revisions to our manuscript as follows:

1. **Inclusion of additional methods:** We have extended the benchmarking analysis to include three additional computational approaches—CARD, cell2location, and STRIDE—In addition to the previously evaluated methods (Tangram, SpaGE, and CIBERSORTx). These additions provide a broader comparison of state-of-the-art methodologies and ensure a more comprehensive evaluation of SEU-TCA's performance.
2. **Selection of key metrics:** Following your suggestion and the benchmarking strategy demonstrated in the SPOT-light paper (Elosua et al., 2021, NAR), we have expanded our evaluation metrics to provide a more comprehensive assessment of SEU-TCA's performance. Specifically, we now assess performance using Accuracy (ACC), F1 score, Sensitivity, and Specificity, which collectively provide a balanced view of the methods' predictive capabilities. Additionally, we have incorporated the Adjusted Rand Index (ARI) to evaluate the consistency between predicted and ground truth distributions of cell types, offering further insights into the spatial accuracy of each method.
3. **Visualization of results:** To better highlight the benchmarking results, we have reorganized the presentation of these analyses. The key comparisons, including the results for ACC, F1 score, Sensitivity, Specificity, and ARI, have been included as part of **Fig. 2** in the main manuscript, with supplementary details provided in **Fig. S1–S5**. This restructuring aligns with your recommendation to show these benchmarks prominently as a main figure.

Through these revisions, we have demonstrated that SEU-TCA consistently outperforms or matches other methods across all datasets and metrics, particularly excelling in distinguishing complex spatial and pathological architectures. We hope these enhancements address your concerns and further strengthen the manuscript’s contribution.

- It is not clear what are the computational requirements of SEU-TCA. The authors should provide computational metrics such as runtime with varying numbers of CPUs and RAM memory consumption, comparing these with similar algorithms. This would provide a clearer understanding of SEU-TCA’s performance relative to existing methods and its feasibility to run without the need for high-performance computing.

Thank you for your insightful comment regarding the computational requirements of SEU-TCA. To address this, we conducted a detailed evaluation of SEU-TCA’s computational efficiency and compared it with other methods.

We examined the runtime of SEU-TCA and the competing methods on the largest dataset in our study—the human heart dataset. All methods were evaluated under the same computational conditions to ensure a fair comparison. The results demonstrate that SEU-TCA achieves a faster runtime compared to most other methods, demonstrating its computational efficiency even for large datasets.

We also evaluated the memory usage during the analysis of the human heart dataset. SEU-TCA exhibited relatively low memory consumption compared to other methods such as Tangram, STRIDE, and CIBERSORTx, demonstrating its efficient resource utilization. The computational performance results, including both runtime and memory consumption, are summarized and presented in **Fig. S3** of the **Additional file 1**.

- In order to prove its general applicability, the authors should test the framework across at least 2-3 other datasets. Examples of available datasets are the Visium datasets on the 10x Genomics website, such as the Adult Mouse Brain Coronal Section (<https://www.10xgenomics.com/datasets/adult-mouse-brain-coronal-section-visium-ff-1-standard>) or pathological contexts like the Human Lung Cancer dataset (<https://www.10xgenomics.com/datasets/human-lung-cancer-11-mm-capture-area-ffpe-2->

standard). These datasets would test not only the applicability of SEU-TCA in different physiological and pathological conditions but also its performance across fresh-frozen and formalin-fixed paraffin-embedded data. Applying SEU-TCA to datasets with diverse characteristics would make clear the applicability of the method.

Thank you for your valuable suggestion regarding the evaluation of SEU-TCA's applicability across diverse datasets. To address this, we have expanded our analysis to include two additional datasets with distinct characteristics:

Mouse olfactory bulb dataset: The mouse olfactory bulb dataset represents a physiologically relevant context, with well-defined anatomical layers including the granule cell layer (GCL), mitral cell layer (MCL), glomerular layer (GL), and nerve layer (ONL). SEU-TCA accurately captured the layered structure and provided superior predictions compared to other methods. This demonstrates SEU-TCA's robustness in reconstructing spatially structured tissues under physiological conditions.

Human pancreatic ductal adenocarcinoma dataset: The pancreatic ductal adenocarcinoma dataset provides a pathological context, containing regions with distinct boundaries, such as ductal cells, acinar cells, stromal regions, and cancer regions. SEU-TCA successfully delineated these regions with high accuracy, resolving complex spatial architectures and demonstrating its suitability to pathological tissues.

The results of these additional analyses are presented in Fig. S4-S5, further supporting SEU-TCA’s utility across diverse datasets and highlighting the generalizability and adaptability of SEU-TCA.

- In page 8 the authors mention “SEU-TCA demonstrated the highest concordance between the prediction and the ground truth”. I do not think the authors have a ground truth to compare their results. Having a real ground truth, for example utilizing a single-cell spatial dataset (performing pseudobulk in spots), or simulated data (as in Kleshchevnikov et al 2022, Nature Biotech) would be crucial to validate the performance of SEU-TCA. Plenty of datasets are available on the 10x Genomics website, such as the Mouse Lung Fresh Frozen (<https://www.10xgenomics.com/datasets/visium-hd-cytassist-gene-expression-mouse-lung-fresh-frozen>) or the Human Lung Cancer dataset (<https://www.10xgenomics.com/datasets/visium-hd-cytassist-gene-expression-human-lung-cancer-fixed-frozen>).

Thank you for pointing out the need for a more rigorous validation using datasets with clear ground truth. We agree that the terminology in our earlier description of the mouse gastrulation dataset results was inaccurate, and we have revised the manuscript to address this issue. Specifically, we have replaced references to “ground truth” with “reference annotations” to better reflect the nature of the comparison.

To address your suggestion, we performed additional analyses using the human heart dataset generated by MERFISH. For this dataset, we constructed pseudo-bulk spatial transcriptomics (ST) data by dividing the tissue into a grid of pseudo-spots. The dominant cell type in each pseudo-spot was determined based on the highest cell-type proportion, serving as the ground truth. We then compared the SEU-TCA predictions for these pseudo-spots against the defined ground truth using multiple evaluation metrics.

This approach ensures that the performance of SEU-TCA is benchmarked against a dataset with a robustly defined ground truth. The results demonstrate that SEU-TCA achieves high concordance with the pseudo-spot ground truth and outperforms other methods in most metrics. These findings are presented in Fig. 2 and Fig. S1 of the revised manuscript.

B

SEU-TCA	Tangram	STRIDE	CARD
  [x] ACC:0.84 [x] ARI:0.64 [x] PCC:0.80 	  ACC:0.84 ARI:0.49 PCC:0.73 	  ACC:0.78 ARI:0.40 PCC:NA 	  ACC:0.76 ARI:0.40 PCC:NA 
	SpaGE   ACC:0.77 ARI:0.52 PCC:0.80 	Cell2location   ACC:0.70 ARI:0.43 PCC:NA 	CIBERSORTx   ACC:0.58 ARI:0.09 PCC:NA 

- Although the authors state that their code is available in Github, as of 4th December the code is not available in the provided github page.

We agree that the importance of detailed documentation in helping researchers to apply SEU-TCA to various spatial datasets. To address this, we have updated the GitHub repository with a more thoroughly commented version of the SEU-TCA notebook (<https://github.com/LinluoLab/SEU-TCA/blob/master/README.md>), explaining the theoretical background, parameter settings, and practical implementation details. Additionally, we have revised the user manual, including a step-by-step guide, kernel parameter optimization strategies (e.g., γ for the RBF kernel), and examples of applying SEU-TCA to different datasets. A quick start guide has also been provided to help users get started easily. These updated resources are now available at <https://github.com/LinluoLab/SEU-TCA>. We hope these improvements address your concerns and support the research community in applying SEU-TCA effectively.

Minor Points

- In the last paragraph of the conclusions, the authors state that “These experiments demonstrate that SEU-TCA is capable of precisely identifying factors with key developmental regulatory functions”. I think this is misleading. SEU-TCA itself does not perform gene regulatory network analyses; rather, this analysis is conducted using the SCENIC algorithm. Clarifying this distinction would enhance the accuracy of the manuscript.

We thank the reviewer for pointing out this important distinction. We acknowledge that the current wording may imply that SEU-TCA directly performs gene regulatory network analyses, which is not accurate. In fact, SEU-TCA enables spatially informed analyses, while SCENIC is specifically used to identify regulatory factors. To clarify this, we have revised the last paragraph of the conclusions section. The revised sentence now reads:

“In summary, SEU-TCA serves as a powerful tool for integrating spatial and transcriptomic data, providing a foundational framework for studying development and disease.”

We appreciate the reviewer’s suggestion, which has helped us enhance the accuracy and clarity of our conclusions.

- In page 14 the phrase “We combined SEU-TCA with the single-cell lineage tracing algorithm, Waddington-Optimal-Transport (WOT)” is not accurate, as this is not a single-cell lineage tracing algorithm, but a differentiation trajectory inference. This should be clarified.

We thank the reviewer for pointing out this inaccuracy. We agree that Waddington-Optimal-Transport (WOT) is not a single-cell lineage tracing algorithm but rather a differentiation trajectory inference method. We have revised the description in the manuscript to reflect this clarification. The revised sentence now reads:

“In our previous study, we employed Waddington-Optimal-Transport (WOT) analysis [58], a differentiation trajectory inference algorithm, to uncover the transcriptional trajectories and epigenetic determinants that specify early cardiac lineages from mesoderm [59].”

We appreciate the reviewer’s attention to detail, which has helped improve the accuracy of our manuscript.

- Although is clear this method is thought to be used in multi-cell spot-based technologies, the authors should at least mention the new emerging technologies that are true spatial single cell.

Thank you for the suggestion. In the revised discussion, we have addressed this point by including the following:

“Emerging single-cell spatial transcriptomics technologies, such as MERFISH [82] and Stereo-seq [83], offer high-resolution insights but they are often constrained by low throughput or prohibitive costs, limiting their feasibility for large-scale studies. In contrast, SEU-TCA leverages widely available multi-cell spot-based data to provide an effective computational framework for spatial transcriptomics analysis. This highlights its importance in bridging the gap between scalability and resolution in current spatial transcriptomics research.”

Second round of review

Reviewer 1

All my points have been addressed, and I believe the paper is now of higher quality. I appreciate that the authors took the time and effort to strengthen the conclusions by adding the missing quantifications, present more of their data, and rephrase their conclusion to convey a more accurate and realistic message.

Reviewer 2

In this revised version the authors addressed all my comments after a thorough revision.

Regarding the major points I raised, I appreciate the authors performed a more in-depth benchmarking of their method, showing that it outperforms or at least performs equally good than other deconvolution methods, with the advantage that this method has a better computational performance in runtime and memory consumption. Also, the authors analysed additional datasets showing the general applicability of their method.

I also appreciate the authors attended to my suggestions to improve the Github page and clarifying some minor points of the text.

We sincerely thank the reviewers for their constructive feedback and positive assessment. Their thoughtful comments greatly contributed to improving the clarity, rigor, and overall quality of our manuscript.